# DIFFUSIONNFT: ONLINE DIFFUSION REINFORCEMENT WITH FORWARD PROCESS

**Kaiwen Zheng**[1,2,*]  **Huayu Chen**[1,2,*]  **Haotian Ye**[2,3]  **Haoxiang Wang**[2]  **Qinsheng Zhang**[2]

**Kai Jiang**[1]  **Hang Su**[1]  **Stefano Ermon**[3]  **Jun Zhu**[1,†]  **Ming-Yu Liu**[2]

[1]Dept. of Comp. Sci. & Tech., BNRist Center, THU-Bosch ML Center, AI Institute, Tsinghua
[2]NVIDIA  [3]Stanford University
https://research.nvidia.com/labs/dir/DiffusionNFT

## ABSTRACT

Online reinforcement learning (RL) has been central to post-training language models, but its extension to diffusion models remains challenging due to intractable likelihoods. Recent works discretize the reverse sampling process to enable GRPO-style training, yet they inherit fundamental drawbacks, including solver restrictions, forward–reverse inconsistency, and complicated integration with classifier-free guidance (CFG). We introduce Diffusion Negative-aware Fine-Tuning (DiffusionNFT), a new online RL paradigm that optimizes diffusion models directly on the forward process via flow matching. DiffusionNFT contrasts positive and negative generations to define an implicit policy improvement direction, naturally incorporating reinforcement signals into the supervised learning objective. This formulation enables training with arbitrary black-box solvers, eliminates the need for likelihood estimation, and requires only clean images rather than sampling trajectories for policy optimization. DiffusionNFT is up to $25\times$ more efficient than FlowGRPO in head-to-head comparisons, while being CFG-free. For instance, DiffusionNFT improves the GenEval score from 0.24 to 0.98 within 1k steps, while FlowGRPO achieves 0.95 with over 5k steps and additional CFG employment. By leveraging multiple reward models, DiffusionNFT significantly boosts the performance of SD3.5-Medium in every benchmark tested.

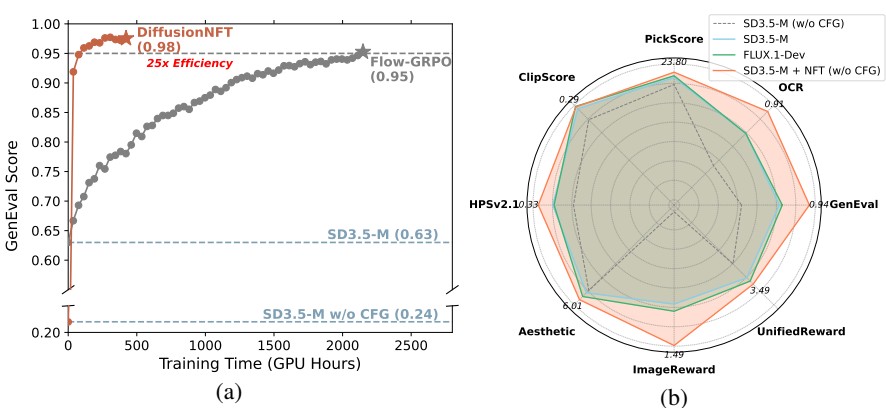

Figure 1: Performance of DiffusionNFT. **(a)** Head-to-head comparison with FlowGRPO on the GenEval task. **(b)** By employing multiple reward models, DiffusionNFT significantly boosts the performance of SD3.5-Medium in every benchmark tested, while being fully CFG-free.

## 1 INTRODUCTION

Online Reinforcement Learning (RL) has been pivotal in the post-training of LLMs, driving recent advances in LLMs' alignment and reasoning abilities (Achiam et al., 2023; Guo et al., 2025). How-

[*]Equal Contribution  [†] Corresponding Author

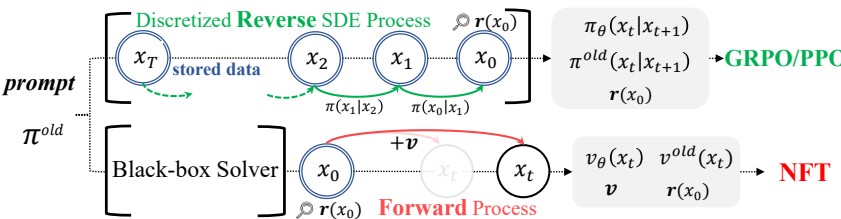

Figure 2: Comparison between Forward-Process RL (NFT) and Reverse-Process RL (GRPO). NFT allows using any solvers and does not require storing the whole sampling trajectory for optimization.

ever, replicating similar success for diffusion models in visual generation is not straightforward. Policy Gradient algorithms assume that model likelihoods are exactly computable. This assumption holds for autoregressive models, but is inherently violated by diffusion models, where likelihoods can only be approximated via costly probabilistic ODE or variational bounds of SDE (Song et al., 2021). Recent works circumvent this barrier by discretizing the reverse sampling process, reframing diffusion generation as a multi-step decision-making problem (Black et al., 2023). This makes transitions between adjacent steps tractable Gaussians, enabling direct application of existing RL algorithms like GRPO to the diffusion domain (Xue et al., 2025; Liu et al., 2025).

Despite promising efforts made, we argue that GRPO-style diffusion reinforcement still faces fundamental limitations: (1) Forward inconsistency. Focusing solely on the reverse sampling process breaks adherence to the forward diffusion process, risking the model degenerating into cascaded Gaussians. (2) Solver restriction. The data collection process relies on first-order SDE samplers, precluding the full utilization of ODE or high-order solvers that are default to flow models and advantageous for generation efficiency. (3) Complicated CFG integration. Diffusion models heavily rely on Classifier-Free Guidance (CFG) (Ho & Salimans, 2022), which requires training both conditional and unconditional models. Current RL practices typically incorporate CFG in post-training, leading to a complicated and inefficient two-model optimization scheme.

We aim to disentangle data collection, remove solver restriction, and maintain consistency with standard supervised pretraining in diffusion RL. As a diffusion policy admits a single forward (noising) process but multiple reverse (denoising) processes (e.g., different samplers), a natural question is:

*Can diffusion reinforcement be performed on the forward process instead of the reverse?*

This paper proposes a novel online RL paradigm named Diffusion Negative-aware FineTuning (DiffusionNFT). Instead of building upon the conventional policy gradient framework, DiffusionNFT directly performs policy optimization on the forward diffusion process through the flow matching objective. Intuitively, it defines a contrastive improvement direction between two implicit policies learned on "positive" and "negative" generated samples split by reward signals, and optimizes toward the positive policy without modifying the sampling process.

The forward-process RL formulation provides several practical benefits (Figure 2). First, DiffusionNFT allows data collection with arbitrary black-box solvers, rather than relying on first-order SDE samplers. Second, it eliminates the need to store entire sampling trajectories, requiring only clean images for policy optimization. Third, it is fully compatible with standard diffusion training, requiring minimal modifications to existing codebases. Finally, it is a native off-policy algorithm, naturally allowing decoupled training and sampling policies without importance sampling.

We evaluate DiffusionNFT by post-training SD3.5-Medium (Esser et al., 2024) on multiple reward models. The entire training process deliberately operates in a CFG-free setting. Although this results in a significantly lower initialization performance, we find DiffusionNFT substantially improves performance across both in-domain and out-of-domain rewards, rapidly outperforming CFG and the GRPO baseline. We also conduct head-to-head comparisons against FlowGRPO in single-reward settings. Across four tasks tested, DiffusionNFT consistently exhibits $3\times$ to $25\times$ efficiency and achieves better final scores. For instance, it improves the GenEval score from 0.24 to 0.98 within 1k steps, while FlowGRPO achieves only 0.95 with over 5k steps and additional CFG employment.

DiffusionNFT is a direct RL alternative to conventional Policy Gradient methods, introducing the Negative-aware FineTuning (NFT) paradigm (Chen et al., 2025c) into the diffusion domain. Grounded in a supervised learning foundation, we believe this paradigm offers a valid path toward a general, unified, and native off-policy RL recipe across various modalities.

## 2 BACKGROUND

### 2.1 DIFFUSION AND FLOW MODELS

Diffusion models (Ho et al., 2020; Song et al., 2020b) learn continuous data distributions by gradually perturbing clean data $\boldsymbol{x}_0 \sim \pi_0 = p_{\text{data}}$ with Gaussian noise according to a forward process. Then, data can be generated by learning to reverse this process.

The forward noising process admits a closed-form transition kernel $\pi_{t|0}(\boldsymbol{x}_t|\boldsymbol{x}_0) = \mathcal{N}(\alpha_t \boldsymbol{x}_0, \sigma_t^2 \mathbf{I})$ with a specific *noise schedule* $\alpha_t, \sigma_t$, enabling reparameterization as

$$\boldsymbol{x}_t = \alpha_t \boldsymbol{x}_0 + \sigma_t \boldsymbol{\epsilon}, \boldsymbol{\epsilon} \sim \mathcal{N}(\mathbf{0}, \mathbf{I}).$$

One way to learn diffusion models is to adopt the velocity parameterization $\boldsymbol{v}_\theta(\boldsymbol{x}_t, t)$ (Zheng et al., 2023b), which predicts the tangent of the trajectory, trained by minimizing

$$\mathbb{E}_{t, \boldsymbol{x}_0 \sim \pi_0, \boldsymbol{\epsilon} \sim \mathcal{N}(\mathbf{0}, \mathbf{I})}[w(t)\|\boldsymbol{v}_\theta(\boldsymbol{x}_t, t) - \boldsymbol{v}\|_2^2], \tag{1}$$

where the target velocity $\boldsymbol{v}$ is defined by the schedule's time derivatives as $\boldsymbol{v} = \dot{\alpha}_t \boldsymbol{x}_0 + \dot{\sigma}_t \boldsymbol{\epsilon}$ under the notation $\dot{f}_t := \mathrm{d} f_t / \mathrm{d}t$, and $w(t)$ is some weighting function. Reverse sampling typically follows the ODE form (Song et al., 2020b) of the diffusion model, which is reduced to $\frac{\mathrm{d}\boldsymbol{x}_t}{\mathrm{d}t} = \boldsymbol{v}_\theta(\boldsymbol{x}_t, t)$ using $\boldsymbol{v}_\theta$. This formulation is known as flow matching (Lipman et al., 2022), where simple Euler discretization serves as an effective ODE solver, equivalent to DDIM (Song et al., 2020a).

Rectified flow (Liu et al., 2022) can be considered as a special case of the above-discussed diffusion models, where $\alpha_t = 1 - t, \sigma_t = t$, which simplifies the velocity target to $\boldsymbol{v} = \boldsymbol{\epsilon} - \boldsymbol{x}_0$.

### 2.2 POLICY GRADIENT ALGORITHMS FOR DIFFUSION MODELS

In order to apply Policy Gradient algorithms such as PPO (Schulman et al., 2017) or GRPO (Shao et al., 2024) to diffusion models, recent works (Black et al., 2023; Fan et al., 2023; Liu et al., 2025; Xue et al., 2025) formulate the diffusion sampling as a multi-step Markov Decision Process (MDP). This can be achieved by discretizing the reverse sampling process of diffusion models.

While flow models naturally admit simple and efficient sampling through ODE, the lack of stochasticity hinders the application of GRPO. FlowGRPO (Liu et al., 2025) addresses this by using the SDE form (Song et al., 2020b) under the velocity parameterization $\boldsymbol{v}_\theta$ (see Appendix B.1):

$$\mathrm{d}\boldsymbol{x}_t = \Big[\boldsymbol{v}_\theta(\boldsymbol{x}_t, t) + \frac{g_t^2}{2t}\big(\boldsymbol{x}_t + (1-t)\boldsymbol{v}_\theta(\boldsymbol{x}_t, t)\big)\Big]\mathrm{d}t + g_t \mathrm{d}\boldsymbol{w}_t \tag{2}$$

where $g_t = a\sqrt{\frac{t}{1-t}}$ controls the level of injected stochasticity. Discretizing it with Euler yields

$$\pi_\theta(\boldsymbol{x}_{t-\Delta t} \mid \boldsymbol{x}_t) = \mathcal{N}\Big(\boldsymbol{x}_t + \Big[\boldsymbol{v}_\theta(\boldsymbol{x}_t, t) + \frac{g_t^2}{2t}(\boldsymbol{x}_t + (1-t)\boldsymbol{v}_\theta(\boldsymbol{x}_t, t))\Big]\Delta t, \ g_t^2 \Delta t \, \mathbf{I}\Big).$$

This makes transition kernels between adjacent steps likelihood tractable Gaussians, enabling the direct application of existing policy gradient algorithms, such as GRPO.

## 3 DIFFUSION REINFORCEMENT VIA NEGATIVE-AWARE FINETUNING

### 3.1 PROBLEM SETUP

**Online RL.** Consider a pretrained diffusion policy $\pi^{\text{old}}$ and prompt datasets $\{\boldsymbol{c}\}$. At each iteration, we sample $K$ images $\boldsymbol{x}_0^{1:K}$ for prompt $\boldsymbol{c}$, and then evaluate each image with a scalar reward function $r \in [0, 1]$, representing its optimality probability $r(\boldsymbol{x}_0, \boldsymbol{c}) := p(\mathbf{o} = 1|\boldsymbol{x}_0, \boldsymbol{c})$ (Levine, 2018).

This optimality serves as a bridge from continuous-valued rewards to a binary partition. Collected data can be randomly split into two imaginary subsets. An image $x_0$ will have a probability $r$ of falling into the positive dataset $\mathcal{D}^+$ and otherwise the negative dataset $\mathcal{D}^-$. Given infinite samples, the underlying distributions of these two subsets are respectively

$$\pi^+(\boldsymbol{x}_0|\boldsymbol{c}) := \pi^{\text{old}}(\boldsymbol{x}_0|\mathbf{o}=1,\boldsymbol{c}) = \frac{p(\mathbf{o}=1|\boldsymbol{x}_0,\boldsymbol{c})\pi^{\text{old}}(\boldsymbol{x}_0|\boldsymbol{c})}{p_{\pi^{\text{old}}}(\mathbf{o}=1|\boldsymbol{c})} = \frac{r(\boldsymbol{x}_0,\boldsymbol{c})}{p_{\pi^{\text{old}}}(\mathbf{o}=1|\boldsymbol{c})}\pi^{\text{old}}(\boldsymbol{x}_0|\boldsymbol{c})$$

$$\pi^-(\boldsymbol{x}_0|\boldsymbol{c}) := \pi^{\text{old}}(\boldsymbol{x}_0|\mathbf{o}=0,\boldsymbol{c}) = \frac{p(\mathbf{o}=0|\boldsymbol{x}_0,\boldsymbol{c})\pi^{\text{old}}(\boldsymbol{x}_0|\boldsymbol{c})}{p_{\pi^{\text{old}}}(\mathbf{o}=0|\boldsymbol{c})} = \frac{1-r(\boldsymbol{x}_0,\boldsymbol{c})}{1-p_{\pi^{\text{old}}}(\mathbf{o}=1|\boldsymbol{c})}\pi^{\text{old}}(\boldsymbol{x}_0|\boldsymbol{c})$$

RL requires performing policy improvement at each iteration. The optimized policy $\pi^*$ satisfies

$$\mathbb{E}_{\pi^*(\cdot|\boldsymbol{c})}r(\boldsymbol{x}_0,\boldsymbol{c}) > \mathbb{E}_{\pi^{\text{old}}(\cdot|\boldsymbol{c})}r(\boldsymbol{x}_0,\boldsymbol{c}) \qquad (\text{denoted as} \quad \pi^* \succ \pi^{\text{old}})$$

**Policy Improvement on Positive Data.** It is easy to prove that $\pi^+ \succ \pi^{\text{old}} \succ \pi^-$ constantly holds, thus a straightforward improvement of $\pi^{\text{old}}$ can be $\pi^* = \pi^+$. To achieve this, previous work (Lee et al., 2023) performs diffusion training solely on $\mathcal{D}^+$, known as Rejection FineTuning (RFT).

Despite the simplicity, RFT cannot effectively leverage negative data in $\mathcal{D}^-$ (Chen et al., 2025c).

**Reinforcement Guidance.** We posit that negative feedback is crucial to policy improvement, especially for diffusion[1]. Rather than treating $\pi^+$ as an optimization *point*, we leverage both negative and positive data to derive an improvement *direction* $\Delta \in \mathbb{R}^n$. The training target is defined as

$$\boldsymbol{v}^*(\boldsymbol{x}_t,\boldsymbol{c},t) := \boldsymbol{v}^{\text{old}}(\boldsymbol{x}_t,\boldsymbol{c},t) + \frac{1}{\beta}\Delta(\boldsymbol{x}_t,\boldsymbol{c},t). \tag{3}$$

where $\boldsymbol{v}$ is the velocity predictor of the diffusion model, $\beta$ is a hyperparameter. This definition formally resembles diffusion guidance such as Classifier-Free Guidance (CFG) (Ho & Salimans, 2022). We term $\Delta(\boldsymbol{x}_t,\boldsymbol{c},t) \in \mathbb{R}^n$ *reinforcement guidance*, and $\frac{1}{\beta} \in \mathbb{R}$ guidance strength.

In Section 3.2, we address two challenges: 1. What is an appropriate form of $\Delta$ that enables policy improvement? 2. How to directly optimize $\boldsymbol{v}_\theta \to \boldsymbol{v}^*$ leveraging collected dataset $\mathcal{D}^+$ and $\mathcal{D}^-$?

## 3.2 NEGATIVE-AWARE DIFFUSION REINFORCEMENT WITH FORWARD PROCESS

In Eq. (3), $\Delta$ corresponds to the distributional shift between an improved policy and the original policy. To formalize this, we first study the distributional difference between $\pi^+ \succ \pi^{\text{old}} \succ \pi^-$.

**Theorem 3.1 (Improvement Direction).** *Consider diffusion models $\boldsymbol{v}^+$, $\boldsymbol{v}^-$, and $\boldsymbol{v}^{old}$ for the policy triplet $\pi^+$, $\pi^-$, and $\pi^{old}$. The directional differences between these models are proportional:*

$$\Delta := [1 - \alpha(\boldsymbol{x}_t)]\,[\boldsymbol{v}^{old}(\boldsymbol{x}_t,\boldsymbol{c},t) - \boldsymbol{v}^-(\boldsymbol{x}_t,\boldsymbol{c},t)] \quad (\textit{Reinforcement Guidance})$$
$$= \alpha(\boldsymbol{x}_t)\,[\boldsymbol{v}^+(\boldsymbol{x}_t,\boldsymbol{c},t) - \boldsymbol{v}^{old}(\boldsymbol{x}_t,\boldsymbol{c},t)]. \tag{4}$$

*where $0 \le \alpha(\boldsymbol{x}_t) \le 1$ is a scalar coefficient:*

$$\alpha(\boldsymbol{x}_t) := \frac{\pi_t^+(\boldsymbol{x}_t|\boldsymbol{c})}{\pi_t^{old}(\boldsymbol{x}_t|\boldsymbol{c})}\mathbb{E}_{\pi^{old}(\boldsymbol{x}_0|\boldsymbol{c})}r(\boldsymbol{x}_0,\boldsymbol{c})$$

Eq. (4) indicates an ideal guidance direction $\Delta$ for improving over $\boldsymbol{v}^{\text{old}}$. With appropriate guidance strength, policy improvement can be guaranteed. For instance, let $\beta = \alpha(\boldsymbol{x}_t)$ in Eq. (3), we have $\boldsymbol{v}^*(\boldsymbol{x}_t,\boldsymbol{c},t) = \boldsymbol{v}^{\text{old}}(\boldsymbol{x}_t,\boldsymbol{c},t) + \frac{1}{\alpha(\boldsymbol{x}_t)}\Delta(\boldsymbol{x}_t,\boldsymbol{c},t) = \boldsymbol{v}^+(\boldsymbol{x}_t,\boldsymbol{c},t)$, such that $\pi^* = \pi^+ \succ \pi^{\text{old}}$ holds. Figure 3 contains an illustration for the improvement direction $\Delta$.

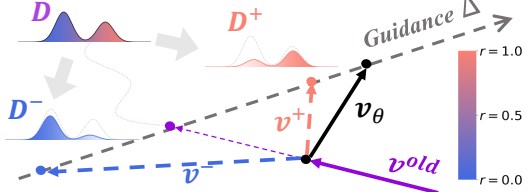

Figure 3: Improvement Direction.

Having defined a valid optimization target $\boldsymbol{v}^*$ with Eq. (3) and (4), we now introduce a training objective that directly optimizes $\boldsymbol{v}_\theta$ towards $\boldsymbol{v}^*$:

---

[1]We find that finetuning only on the positive data leads to collapse (Section 4.4)

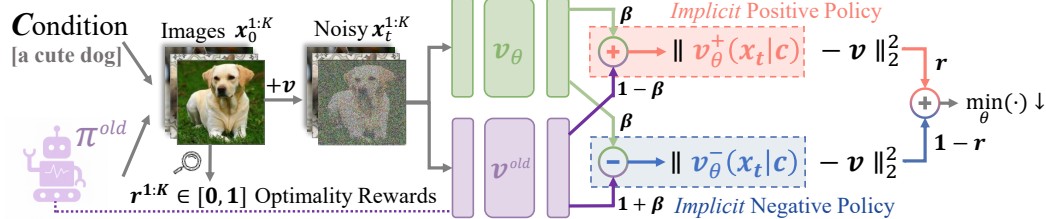

Figure 4: DiffusionNFT jointly optimizes two dual diffusion objectives, on both positive ($r = 1$) and negative ($r = 0$) branches. Rather than training two independent models $\boldsymbol{v}_\theta^+$ and $\boldsymbol{v}_\theta^-$, it adopts an implicit parameterization technique that directly optimizes a single target policy $\boldsymbol{v}_\theta$.

**Theorem 3.2** (**Policy Optimization**). *Consider the training objective:*

$$\mathcal{L}(\theta) = \mathbb{E}_{\boldsymbol{c}, \pi^{old}(\boldsymbol{x}_0|\boldsymbol{c}), t}\, r\|\boldsymbol{v}_\theta^+(\boldsymbol{x}_t, \boldsymbol{c}, t) - \boldsymbol{v}\|_2^2 + (1-r)\|\boldsymbol{v}_\theta^-(\boldsymbol{x}_t, \boldsymbol{c}, t) - \boldsymbol{v}\|_2^2, \quad (5)$$

*where* $\quad \boldsymbol{v}_\theta^+(\boldsymbol{x}_t, \boldsymbol{c}, t) := (1-\beta)\boldsymbol{v}^{old}(\boldsymbol{x}_t, \boldsymbol{c}, t) + \beta\boldsymbol{v}_\theta(\boldsymbol{x}_t, \boldsymbol{c}, t), \quad$ *(Implicit positive policy)*

*and* $\quad \boldsymbol{v}_\theta^-(\boldsymbol{x}_t, \boldsymbol{c}, t) := (1+\beta)\boldsymbol{v}^{old}(\boldsymbol{x}_t, \boldsymbol{c}, t) - \beta\boldsymbol{v}_\theta(\boldsymbol{x}_t, \boldsymbol{c}, t). \quad$ *(Implicit negative policy)*

*Given unlimited data and model capacity, the optimal solution of Eq. (5) satisfies*

$$\boldsymbol{v}_{\theta^*}(\boldsymbol{x}_t, \boldsymbol{c}, t) = \boldsymbol{v}^{old}(\boldsymbol{x}_t, \boldsymbol{c}, t) + \frac{2}{\beta}\Delta(\boldsymbol{x}_t, \boldsymbol{c}, t). \quad (6)$$

Theorem 3.2 presents a new off-policy RL paradigm (Figure 4). Instead of applying Policy Gradient, it adopts supervised learning (SL) objectives, but additionally trains on online negative data $\mathcal{D}^-$. This renders the algorithm highly versatile, compatible with existing SL methods. We term our method *Diffusion Negative-aware FineTuning* (**DiffusionNFT**), highlighting its negative-aware SL nature and conceptual similarity to parallel algorithm NFT in language models (Chen et al., 2025c).

Below, we discuss several distinctive advantages of DiffusionNFT.

**1. Forward Consistency.** In contrast to policy gradient methods (e.g., FlowGRPO), which formulated RL on the reverse diffusion process, DiffusionNFT defines a typical diffusion loss on the forward process. This preserves what we term *forward consistency*—the adherence of the diffusion model's underlying probability density to the Fokker-Planck equation (Øksendal, 2003; Song et al., 2020b), ensuring that the learned model corresponds to a valid forward process (i.e., $\boldsymbol{x}_t$ are correctly coupled with $\boldsymbol{x}_0$ through a joint distribution $\pi_\theta(\boldsymbol{x}_t, \boldsymbol{x}_0) = \pi_\theta(\boldsymbol{x}_0)\pi_{t|0}(\boldsymbol{x}_t|\boldsymbol{x}_0)$).

**2. Solver Flexibility.** DiffusionNFT fully decouples policy training and data sampling. This enables the full utilization of any black-box solvers throughout sampling, rather than relying on first-order SDE samplers. It also eliminates the need to store the entire sampling trajectory during data collection, requiring only clean images with their associated rewards for training.

**3. Implicit Guidance Integration.** Intuitively, DiffusionNFT defines a guidance direction $\Delta$ and apply such guidance to the old policy $\boldsymbol{v}^{old}$ (Eq. (6)). However, instead of learning a separate guidance model $\Delta_\theta$ and employing guided sampling, it adopts an implicit parameterization technique that enables direct integration of reinforcement guidance into the learned policy. This technique, inspired by recent advances in guidance-free training (Chen et al., 2025a), allows us to perform RL continuously on a single policy model, which is crucial to online reinforcement.

**4. Likelihood-Free Formulation.** Previous diffusion RL methods are fundamentally constrained by their reliance on likelihood approximation. Whether approximating the marginal data likelihood with variational bounds and applying Jensen's inequality to reduce loss computation cost (Wallace et al., 2024), or discretizing the reverse process to estimate sequence likelihood (Black et al., 2023), they inevitably introduce systematic estimation bias into diffusion post-training. In contrast, DiffusionNFT is inherently likelihood-free, bypassing such compromises.

### 3.3 PRACTICAL IMPLEMENTATION

We provide DiffusionNFT pseudo code in Algorithm 1. Below, we elaborate on key design choices.

---

**Algorithm 1** Diffusion Negative-aware FineTuning (**DiffusionNFT**)

---

**Require:** Pretrained diffusion policy $v^{\text{ref}}$, raw reward function $r^{\text{raw}}(\cdot) \in \mathbb{R}$, prompt dataset $\{c\}$.
**Initialize:** Data collection policy $v^{\text{old}} \leftarrow v^{\text{ref}}$, training policy $v_\theta \leftarrow v^{\text{ref}}$, data buffer $\mathcal{D} \leftarrow \emptyset$.

1: **for** each iteration $i$ **do**
2:      **for** each sampled prompt $c$ **do**            *// Rollout Step, Data Collection*
3:          Collect $K$ clean images $x_0^{1:K}$, and evaluate their rewards $\{r^{\text{raw}}\}^{1:K}$.
4:          Normalize raw rewards in group: $r^{\text{norm}} := r^{\text{raw}} - \texttt{mean}(\{r^{\text{raw}}\}^{1:K})$.
5:          Define optimality probability $r = 0.5 + 0.5 * \texttt{clip}\{r^{\text{norm}}/Z_c, -1, 1\}$.
6:          $\mathcal{D} \leftarrow \{c, \ x_0^{1:K}, r^{1:K} \in [0,1]\}$.
7:      **end for**
8:      **for** each mini batch $\{c, x_0, r\} \in \mathcal{D}$ **do**          *// Gradient Step, Policy Optimization*
9:          Forward diffusion process: $x_t = \alpha_t x_0 + \sigma_t \epsilon$;   $v = \dot{\alpha}_t x_0 + \dot{\sigma}_t \epsilon$.
10:         Implicit positive velocity: $v_\theta^+(x_t, c, t) := (1 - \beta)v^{\text{old}}(x_t, c, t) + \beta v_\theta(x_t, c, t)$.
11:         Implicit negative velocity: $v_\theta^-(x_t, c, t) := (1 + \beta)v^{\text{old}}(x_t, c, t) - \beta v_\theta(x_t, c, t)$.
12:         $\theta \leftarrow \theta - \lambda \nabla_\theta \left[ r\|v_\theta^+(x_t, c, t) - v\|_2^2 + (1-r)\|v_\theta^-(x_t, c, t) - v\|_2^2 \right].$     (Eq. (5))
13:      **end for**
14:      Update data collection policy $\theta^{\text{old}} \leftarrow \eta_i \theta^{\text{old}} + (1 - \eta_i)\theta$, and clear buffer $\mathcal{D} \leftarrow \emptyset$.    *// Online Update*
15: **end for**

**Output:** $v_\theta$

---

**Optimality Reward.** In most visual reinforcement settings, rewards manifest as unconstrained continuous scalars rather than binary optimality signals. Motivated by existing GRPO practices (Shao et al., 2024; Liu et al., 2025; Xue et al., 2025), we first transform the raw reward $r^{\text{raw}}$ into $r \in [0, 1]$ which represents the optimality probability:

$$r(x_0, c) := \frac{1}{2} + \frac{1}{2}\texttt{clip}\left[\frac{r^{\text{raw}}(x_0, c) - \mathbb{E}_{\pi^{\text{old}}(\cdot|c)}r^{\text{raw}}(x_0, c)}{Z_c}, -1, 1\right].$$

$Z_c > 0$ is some normalizing factor, which could take the form of a global reward $\texttt{std}$. We sample $K$ images for each prompt $c$ during data collection, so the average reward $\mathbb{E}_{\pi^{\text{old}}(\cdot|c)}r^{\text{raw}}(x_0, c)$ for each prompt can be estimated.

**Soft Update of Sampling Policy.** The off-policy nature of DiffusionNFT decouples the sampling policy $\pi^{\text{old}}$ from the training policy $\pi_\theta$. This obviates the need for a "hard" update ($\pi^{\text{old}} \leftarrow \pi^\theta$) after each iteration. Instead, we leverage this property to employ a "soft" EMA update:

$$\theta^{\text{old}} \leftarrow \eta_i \theta^{\text{old}} + (1 - \eta_i)\theta$$

where $i$ is the iteration number. The parameter $\eta$ governs a trade-off between learning speed and stability. A strictly on-policy scheme ($\eta = 0$) yields rapid initial progress but is prone to severe instability, leading to catastrophic collapse. Conversely, a nearly offline approach ($\eta \to 1$) is robustly stable but suffers from impractically slow convergence (Figure 8).

**Adaptive Loss Weighting.** Typical diffusion loss includes a time-dependent weighting $w(t)$ (Eq. (1)). Instead of manual tuning, we adopt an adaptive weighting scheme. The velocity predictor $v_\theta$ can be equivalently transformed into $x_0$ predictor, denoted as $x_\theta$ (e.g., $x_\theta = x_t - tv_\theta$ under rectified flow schedule). We replace the weighting with a form of self-normalized $x_0$ regression, motivated by the diffusion distillation method DMD (Yin et al., 2024):

$$w(t)\|v_\theta(x_t, c, t) - v\|_2^2 \leftarrow \frac{\|x_\theta(x_t, c, t) - x_0\|_2^2}{\texttt{sg}(\texttt{mean}(\texttt{abs}(x_\theta(x_t, c, t) - x_0)))}$$

where $\texttt{sg}$ is the stop-gradient operator. We find it typically leads to faster training (Figure 9).

**CFG-Free Optimization.** Classifier-Free Guidance (CFG) (Ho & Salimans, 2022) is a default technique to enhance generation quality at inference time, yet it complicates post-training and reduces efficiency. Conceptually, we interpret CFG as an offline form of reinforcement guidance (Eq. (4)), where conditional and unconditional models correspond to positive and negative signals. With this understanding, we discard CFG in our algorithm design, and the policy is initialized solely by the conditional model. Despite this seemingly poor initialization, we observe that performance surges and quickly surpasses the CFG baseline (Figure 1). This suggests that the functionality of CFG can be effectively learned or substituted through RL post-training, echoing recent studies that achieve strong performance without CFG through post-training (Chen et al., 2025b;a; Zheng et al., 2025).

Table 1: **Evaluation Results.** Gray-colored: In-domain reward. [†] Evaluated on official checkpoints. [‡] Evaluated under 1024×1024 resolution. **Bold**: best; Underline: second best.

| Model | #Iter | Rule-Based | | Model-Based | | | | | |
|---|---|---|---|---|---|---|---|---|---|
| | | GenEval | OCR | PickScore | ClipScore | HPSv2.1 | Aesthetic | ImgRwd | UniRwd |
| SD-XL[‡] | — | 0.55 | 0.14 | 22.42 | 0.287 | 0.280 | 5.60 | 0.76 | 2.93 |
| SD3.5-L[‡] | — | 0.71 | 0.68 | 22.91 | 0.289 | 0.288 | 5.50 | 0.96 | 3.25 |
| FLUX.1-Dev | — | 0.66 | 0.59 | 22.84 | 0.295 | 0.274 | 5.71 | 0.96 | 3.27 |
| SD3.5-M (w/o CFG) | — | 0.24 | 0.12 | 20.51 | 0.237 | 0.204 | 5.13 | -0.58 | 2.02 |
| + CFG | — | 0.63 | 0.59 | 22.34 | 0.285 | 0.279 | 5.36 | 0.85 | 3.03 |
| + FlowGRPO[†] | >5k | **0.95** | 0.66 | 22.51 | **0.293** | 0.274 | 5.32 | 1.06 | 3.18 |
| | 2k | 0.66 | **0.92** | 22.41 | 0.290 | 0.280 | 5.32 | 0.95 | 3.15 |
| | 4k | 0.54 | 0.68 | 23.50 | 0.280 | 0.316 | 5.90 | 1.29 | 3.37 |
| + Ours | 1.7k | 0.94 | 0.91 | **23.80** | **0.293** | **0.331** | **6.01** | **1.49** | **3.49** |

## 4 EXPERIMENTS

We demonstrate the potential of DiffusionNFT through three perspectives: (1) multi-reward joint training for strong CFG-free performance, (2) head-to-head comparison with FlowGRPO on single rewards, and (3) ablation studies on key design choices.

### 4.1 EXPERIMENTAL SETUP

Our experiments are based on `SD3.5-Medium` (Esser et al., 2024) at 512×512 resolution, with most settings aligned with FlowGRPO (Liu et al., 2025).

**Reward Models.** (1) Rule-based rewards, including `GenEval` (Ghosh et al., 2023) for compositional image generation and `OCR` for visual text rendering, where the partial reward assignment strategies follow FlowGRPO. (2) Model-based rewards, including `PickScore` (Kirstain et al., 2023), `ClipScore` (Hessel et al., 2021), `HPSv2.1` (Wu et al., 2023), `Aesthetics` (Schuhmann, 2022), `ImageReward` (Xu et al., 2023) and `UnifiedReward` (Wang et al., 2025), which measure image quality, image-text alignment and human preference.

**Prompt Datasets.** For `GenEval` and `OCR`, we use the corresponding training and test sets from FlowGRPO. For other rewards, we train on `Pick-a-Pic` (Kirstain et al., 2023) and evaluate on `DrawBench` (Saharia et al., 2022).

**Training and Evaluation.** We finetune with LoRA ($\alpha = 64$, $r = 32$). Each epoch consists of 48 groups with group size $G = 24$. We use 10 rollout sampling steps for head-to-head comparison and ablation studies, and 40 steps for best visual quality in multi-reward training. Evaluation is performed with 40-step first-order ODE sampler. Additional details are provided in Appendix C.

### 4.2 MULTI-REWARD JOINT TRAINING

We first assess DiffusionNFT's effectiveness in comprehensively enhancing the base model. Starting from the CFG-free SD3.5-M (2.5B parameters), we jointly optimize five rewards: `GenEval`, `OCR`, `PickScore`, `ClipScore`, and `HPSv2.1`. Since the rewards are based on different prompts, we first train on `Pick-a-Pic` with model-based rewards to strengthen alignment and human preference, followed by rule-based rewards (`GenEval`, `OCR`). Out-of-domain evaluation is conducted on `Aesthetics`, `ImageReward`, and `UnifiedReward`.

As shown in Table 1, our final CFG-free model not only surpasses CFG and matches FlowGRPO (fitted only single rewards) on both in-domain and out-of-domain metrics, but also outperforms CFG-based larger models such as SD3.5-L (8B parameters) and FLUX.1-Dev (12B parameters) (Labs, 2024). Qualitative comparison in Figure 5 demonstrates the superior visual quality of our method.

### 4.3 HEAD-TO-HEAD COMPARISON

We conduct head-to-head comparisons with FlowGRPO on single training rewards. As shown in Figure 1(a) and Figure 6, our method is 3× to 25× more efficient in terms of wall-clock time,

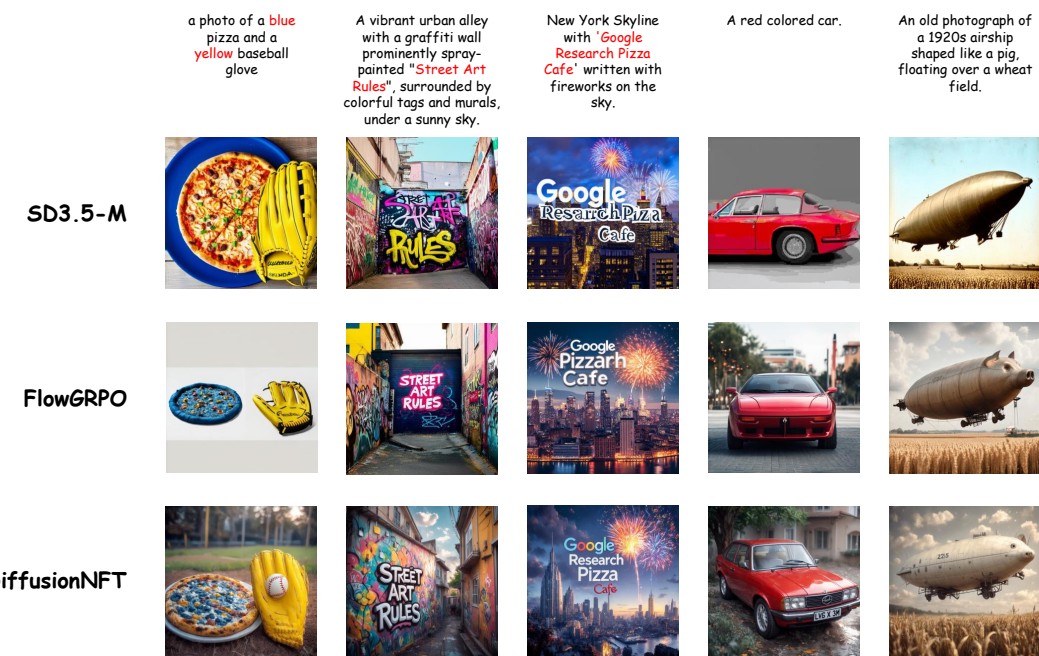

Figure 5: **Qualitative Comparison.** The prompts are taken from `GenEval`, `OCR` and `DrawBench` respectively, where we compare the corresponding FlowGRPO model with our model.

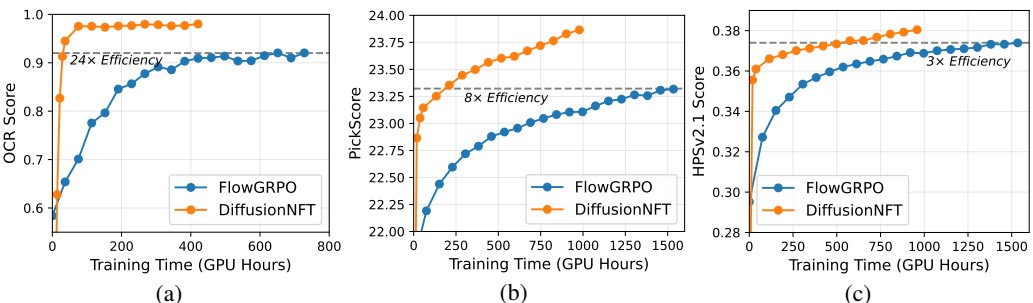

Figure 6: Head-to-head comparison between DiffusionNFT with FlowGRPO on single rewards.

achieving `GenEval` score of 0.98 within only ∼1k iterations. This demonstrates that CFG-free models can rapidly adapt to specific reward environments under our framework.

## 4.4 Ablation Studies

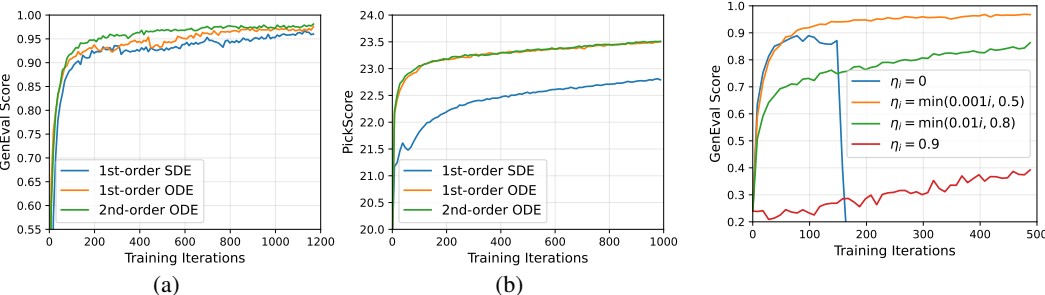

Figure 7: Different diffusion samplers for data collection.

Figure 8: Soft-update strategies.

We analyze the impact of our core design choices:

**Negative Loss.** The negative-aware component is crucial in DiffusionNFT. Without the negative policy loss on $v_\theta^-$, we find rewards collapse almost instantly during online training, highlighting the

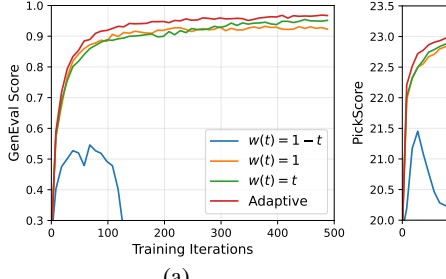 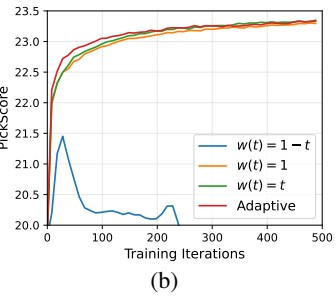 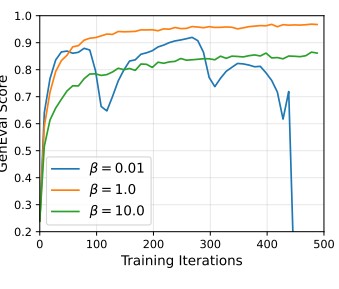

Figure 9: Different time-dependent weighting strategies.    Figure 10: Choices of strength $\beta$.

essential role of negative signals in diffusion RL. This phenomenon is divergent from observations in LLMs, where RFT remains a strong baseline (Xiong et al., 2025; Chen et al., 2025c).

**Diffusion Sampler.** Online samples in DiffusionNFT are used both for reward evaluation and as training data, making quality critical. Figure 7 shows that ODE samplers outperform SDE ones, especially on `PickScore`, which is noise-sensitive. Second-order ODE slightly outperforms first-order on `GenEval`, while being comparable on `PickScore`.

**Adaptive Weighting.** We find stability improves when the flow-matching loss is given higher weight at larger $t$, whereas inverse strategies (e.g., $w(t) = 1 - t$) lead to collapse (Figure 9). Our adaptive schedule consistently matches or exceeds heuristic choices.

**Soft Update.** We compare different $\eta_i$ schedules for the soft update in Figure 8. Fully on-policy ($\eta_i = 0$) accelerates early progress but destabilizes training, while overly off-policy ($\eta = 0.9$) slows convergence. We find that starting with a small $\eta$ and gradually increasing it to a larger value in later stages strikes an effective balance between convergence speed and training stability.

**Guidance Strength.** As shown in Figure 10, the guidance parameter $\beta$ also governs a trade-off between stability and convergence speed. We find that $\beta$ near 1 performs stably and select $\beta$ as 1 or 0.1 (for faster reward increase) in practice.

## 5 RELATED WORK

The transition of RL algorithms from discrete autoregressive (AR) to continuous diffusion models poses a central challenge: the inherent difficulty of diffusion models for computing exact model likelihoods (Song et al., 2021), which are nonetheless crucial for RL (Chen et al., 2023; Liu et al., 2025). To address this challenge, existing efforts include:

*Likelihood-free* methods: (1) Reward Backpropagation (Xu et al., 2023; Prabhudesai et al., 2023; Clark et al., 2023; Prabhudesai et al., 2024) proves highly effective, yet is limited to differentiable rewards and can only tune low-noise timesteps due to memory costs and gradient explosion when unrolling long denoising chains. (2) Reward-Weighted Regression (RWR) (Lee et al., 2023) is an offline finetuning method but lacks a negative policy objective to penalize low-reward generations. (3) Policy Guidance. This includes energy guidance (Janner et al., 2022; Lu et al., 2023) and CFG-style guidance (Frans et al., 2025; Jin et al., 2025). These methods all require combining multiple models for guided sampling, thus complicating online optimization. (4) Score-based RL. These methods try to perform RL directly on the score rather than the likelihood field (Zhu et al., 2025).

*Likelihood-based* methods: (1) Diffusion-DPO (Wallace et al., 2024; Yang et al., 2024; Liang et al., 2024; Yuan et al., 2024; Li et al., 2025a) adapts DPO to diffusion for paired human preference data but requires additional likelihood and loss approximations compared to AR; DDO (Zheng et al., 2025) uses high-quality dataset as positive signals and self-generated samples as negative signals to avoid the requirement of paired data, achieving state-of-the-art CFG-free FIDs in visual generation, while still relying on likelihood approximation for the diffusion case. (2) Policy gradient methods, starting from PPO style (Black et al., 2023; Fan et al., 2023), decompose trajectory likelihoods step by step without considering forward consistency. Recent GRPO extensions (Liu et al., 2025; Xue et al., 2025) prove effective and scalable for diffusion RL, but they couple the training loss with

SDE samplers and face efficiency bottlenecks. MixGRPO (Li et al., 2025b) improves efficiency by mixing SDE and ODE, while issues of coupling and forward inconsistency remain.

## 6 CONCLUSION

We introduce Diffusion Negative-aware FineTuning (DiffusionNFT), a new paradigm for online reinforcement learning of diffusion models that directly operates on the forward process. By formulating policy improvement as a contrast between positive and negative generations, DiffusionNFT integrates reinforcement signals seamlessly into the standard diffusion objective, eliminating the reliance on likelihood estimation and SDE-based reverse process. Empirically, DiffusionNFT demonstrates strong and efficient reward optimization, achieving up to $25\times$ higher efficiency than Flow-GRPO while producing a single model that outperforms CFG baselines across diverse in-domain and out-of-domain rewards. We believe this work represents a step toward unifying supervised and reinforcement learning in diffusion, and highlights the forward process as a promising foundation for scalable, efficient, and theoretically principled diffusion RL.

### THE USE OF LARGE LANGUAGE MODELS (LLMS)

We used large language models (LLMs) solely as a writing assistant for language polishing and improving clarity of presentation. The LLMs were not involved in research ideation, methodological design, experimental execution, or result analysis. All scientific contributions and substantive writing were carried out by the authors.

### ACKNOWLEDGMENTS

This work was supported by Fundamental and Interdisciplinary Disciplines Breakthrough Plan of the Ministry of Education of China (No. JYB2025XDXM101), NSF of China Projects (Nos. 62550004, U25B6003, 92370124, 92248303); Beijing Natural Science Foundation L247011; the High Performance Computing Center, Tsinghua University. J.Z was also supported by the XPlorer Prize.

We thank Cheng Lu, Hanzi Mao, Zekun Hao, Tao Yang, Zhanhao Liang, Shuhuai Ren, Tenglong Ao, Xintao Wang, Haoqi Fan, Jiajun Liang, Yuji Wang, and Hongzhou Zhu for the valuable discussion.

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

## A    PROOF OF THEOREMS

**Lemma A.1** (**Distribution Split**). *Consider the distribution triplet $\pi^+$, $\pi^-$, and $\pi^{old}$, as defined in Section 3.1:*

$$\pi^+(\boldsymbol{x}_0|\boldsymbol{c}) := \pi^{old}(\boldsymbol{x}_0|\mathbf{o}=1,\boldsymbol{c}) = \frac{p(\mathbf{o}=1|\boldsymbol{x}_0,\boldsymbol{c})\pi^{old}(\boldsymbol{x}_0|\boldsymbol{c})}{p_{\pi^{old}}(\mathbf{o}=1|\boldsymbol{c})} = \frac{r(\boldsymbol{x}_0,\boldsymbol{c})}{p_{\pi^{old}}(\mathbf{o}=1|\boldsymbol{c})}\pi^{old}(\boldsymbol{x}_0|\boldsymbol{c}) \quad (7)$$

$$\pi^-(\boldsymbol{x}_0|\boldsymbol{c}) := \pi^{old}(\boldsymbol{x}_0|\mathbf{o}=0,\boldsymbol{c}) = \frac{p(\mathbf{o}=0|\boldsymbol{x}_0,\boldsymbol{c})\pi^{old}(\boldsymbol{x}_0|\boldsymbol{c})}{p_{\pi^{old}}(\mathbf{o}=0|\boldsymbol{c})} = \frac{1-r(\boldsymbol{x}_0,\boldsymbol{c})}{1-p_{\pi^{old}}(\mathbf{o}=1|\boldsymbol{c})}\pi^{old}(\boldsymbol{x}_0|\boldsymbol{c}) \quad (8)$$

*$\pi^{old}(\boldsymbol{x}_0|\boldsymbol{c})$ is as a linear combination between its positive split $\pi^+(\boldsymbol{x}_0|\boldsymbol{c})$ and negative split $\pi^-(\boldsymbol{x}_0|\boldsymbol{c})$:*

$$\pi^{old}(\boldsymbol{x}_0|\boldsymbol{c}) = p_{\pi^{old}}(\mathbf{o}=1|\boldsymbol{c})\pi^+(\boldsymbol{x}_0|\boldsymbol{c}) + [1-p_{\pi^{old}}(\mathbf{o}=1|\boldsymbol{c})]\pi^-(\boldsymbol{x}_0|\boldsymbol{c}) \quad (9)$$

*Proof.* The result follows directly from Eq.(7) and Eq.(8). $\qquad\square$

**Lemma A.2** (**Posterior Split**). *The diffusion posteriors for distribution triplet $\pi^+$, $\pi^-$, and $\pi^{old}$ satisfy:*

$$\pi^{old}(\boldsymbol{x}_0|\boldsymbol{x}_t,\boldsymbol{c}) = \alpha(\boldsymbol{x}_t)\pi^+(\boldsymbol{x}_0|\boldsymbol{x}_t,\boldsymbol{c}) + [1-\alpha(\boldsymbol{x}_t)]\pi^-(\boldsymbol{x}_0|\boldsymbol{x}_t,\boldsymbol{c})$$

$$where \qquad \alpha(\boldsymbol{x}_t) := \frac{\pi_t^+(\boldsymbol{x}_t|\boldsymbol{c})}{\pi_t^{old}(\boldsymbol{x}_t|\boldsymbol{c})}\mathbb{E}_{\pi^{old}(\boldsymbol{x}_0|\boldsymbol{c})}r(\boldsymbol{x}_0,\boldsymbol{c})$$

*Proof.* Leveraging Bayes' Rule:

$$\pi^{old}(\boldsymbol{x}_0|\boldsymbol{c}) = \frac{\pi_t^{old}(\boldsymbol{x}_t|\boldsymbol{c})\pi_{0|t}^{old}(\boldsymbol{x}_0|\boldsymbol{x}_t,\boldsymbol{c})}{\pi(\boldsymbol{x}_t|\boldsymbol{x}_0)}$$

Replacing all distributions in Eq. (9) (Lemma A.1) we get

$$\frac{\pi_t^{old}(\boldsymbol{x}_t|\boldsymbol{c})\pi_{0|t}^{old}(\boldsymbol{x}_0|\boldsymbol{x}_t,\boldsymbol{c})}{\pi(\boldsymbol{x}_t|\boldsymbol{x}_0)} = p_{\pi^{old}}(\mathbf{o}=1|\boldsymbol{c})\frac{\pi_t^+(\boldsymbol{x}_t|\boldsymbol{c})\pi_{0|t}^+(\boldsymbol{x}_0|\boldsymbol{x}_t,\boldsymbol{c})}{\pi(\boldsymbol{x}_t|\boldsymbol{x}_0)}$$

$$+ [1-p_{\pi^{old}}(\mathbf{o}=1|\boldsymbol{c})]\frac{\pi_t^-(\boldsymbol{x}_t|\boldsymbol{c})\pi_{0|t}^-(\boldsymbol{x}_0|\boldsymbol{x}_t,\boldsymbol{c})}{\pi(\boldsymbol{x}_t|\boldsymbol{x}_0)}$$

$$\Rightarrow \pi_{0|t}^{old}(\boldsymbol{x}_0|\boldsymbol{x}_t,\boldsymbol{c}) = p_{\pi^{old}}(\mathbf{o}=1|\boldsymbol{c})\frac{\pi_t^+(\boldsymbol{x}_t|\boldsymbol{c})}{\pi_t^{old}(\boldsymbol{x}_t|\boldsymbol{c})}\pi_{0|t}^+(\boldsymbol{x}_0|\boldsymbol{x}_t,\boldsymbol{c})$$

$$+ [1-p_{\pi^{old}}(\mathbf{o}=1|\boldsymbol{c})]\frac{\pi_t^-(\boldsymbol{x}_t|\boldsymbol{c})}{\pi_t^{old}(\boldsymbol{x}_t|\boldsymbol{c})}\pi_{0|t}^-(\boldsymbol{x}_0|\boldsymbol{x}_t,\boldsymbol{c})$$

Diffuse both sides of Eq. (9), we have

$$\pi_t^{old}(\boldsymbol{x}_t|\boldsymbol{c}) = p_{\pi^{old}}(\mathbf{o}=1|\boldsymbol{c})\pi_t^+(\boldsymbol{x}_t|\boldsymbol{c}) + [1-p_{\pi^{old}}(\mathbf{o}=1|\boldsymbol{c})]\pi_t^-(\boldsymbol{x}_t|\boldsymbol{c})$$

$$p_{\pi^{old}}(\mathbf{o}=1|\boldsymbol{c})\frac{\pi_t^+(\boldsymbol{x}_t|\boldsymbol{c})}{\pi_t^{old}(\boldsymbol{x}_t|\boldsymbol{c})} + [1-p_{\pi^{old}}(\mathbf{o}=1|\boldsymbol{c})]\frac{\pi_t^-(\boldsymbol{x}_t|\boldsymbol{c})}{\pi_t^{old}(\boldsymbol{x}_t|\boldsymbol{c})} = 1$$

Note that

$$p_{\pi^{old}}(\mathbf{o}=1|\boldsymbol{c}) = \mathbb{E}_{\pi^{old}(\boldsymbol{x}_0|\boldsymbol{c})}r(\boldsymbol{x}_0,\boldsymbol{c})$$

We have

$$\pi_{0|t}^{old}(\boldsymbol{x}_0|\boldsymbol{x}_t,\boldsymbol{c}) = \alpha(\boldsymbol{x}_t)\pi_{0|t}^+(\boldsymbol{x}_0|\boldsymbol{x}_t,\boldsymbol{c}) + [1-\alpha(\boldsymbol{x}_t)]\pi_{0|t}^-(\boldsymbol{x}_0|\boldsymbol{x}_t,\boldsymbol{c})$$

$$\square$$

**Theorem A.3** (**Improvement Direction**). *Consider diffusion models $\boldsymbol{v}^+$, $\boldsymbol{v}^-$, and $\boldsymbol{v}^{old}$ for the distribution triplet $\pi^+$, $\pi^-$, and $\pi^{old}$. The directional differences between these models are parallel:*

$$\Delta := [1-\alpha(\boldsymbol{x}_t)]\left[\boldsymbol{v}^{old}(\boldsymbol{x}_t,\boldsymbol{c},t) - \boldsymbol{v}^-(\boldsymbol{x}_t,\boldsymbol{c},t)\right] \quad \textit{(Reinforcement Guidance)}$$

$$= \alpha(\boldsymbol{x}_t)\left[\boldsymbol{v}^+(\boldsymbol{x}_t,\boldsymbol{c},t) - \boldsymbol{v}^{old}(\boldsymbol{x}_t,\boldsymbol{c},t)\right].$$

*where $0 \leq \alpha(\boldsymbol{x}_t) \leq 1$ is a scalar coefficient:*

$$\alpha(\boldsymbol{x}_t) := \frac{\pi_t^+(\boldsymbol{x}_t|\boldsymbol{c})}{\pi_t^{old}(\boldsymbol{x}_t|\boldsymbol{c})}\mathbb{E}_{\pi^{old}(\boldsymbol{x}_0|\boldsymbol{c})}r(\boldsymbol{x}_0,\boldsymbol{c})$$

*Proof.* According to the relationship between the optimal velocity predictor and the posterior mean of $\boldsymbol{x}_0$ (i.e., the optimal $\boldsymbol{x}_0$ predictor) (Zheng et al., 2023b):

$$\boldsymbol{v}^{\text{old}}(\boldsymbol{x}_t, \boldsymbol{c}, t) = a_t \boldsymbol{x}_t + b_t \mathbb{E}_{\pi^{\text{old}}(\boldsymbol{x}_0|\boldsymbol{x}_t, \boldsymbol{c})}[\boldsymbol{x}_0]$$

$$\boldsymbol{v}^+(\boldsymbol{x}_t, \boldsymbol{c}, t) = a_t \boldsymbol{x}_t + b_t \mathbb{E}_{\pi^+(\boldsymbol{x}_0|\boldsymbol{x}_t, \boldsymbol{c})}[\boldsymbol{x}_0]$$

$$\boldsymbol{v}^-(\boldsymbol{x}_t, \boldsymbol{c}, t) = a_t \boldsymbol{x}_t + b_t \mathbb{E}_{\pi^=(\boldsymbol{x}_0|\boldsymbol{x}_t, \boldsymbol{c})}[\boldsymbol{x}_0]$$

where $a_t = \frac{\dot{\sigma}_t}{\sigma_t}, b_t = \dot{\alpha}_t - \frac{\dot{\sigma}_t \alpha_t}{\sigma_t}$. Based on Lemma A.2 we have

$$\boldsymbol{v}^{\text{old}}(\boldsymbol{x}_t, \boldsymbol{c}, t) = \alpha(\boldsymbol{x}_t) \boldsymbol{v}^+(\boldsymbol{x}_t, \boldsymbol{c}, t) + [1 - \alpha(\boldsymbol{x}_t)] \boldsymbol{v}^-(\boldsymbol{x}_t, \boldsymbol{c}, t)$$

Rearranging the equation, we complete the proof. $\qquad\square$

**Theorem A.4** (**Reinforcement Guidance Optimization**). *Consider the training objective:*

$$\mathcal{L}(\theta) = \mathbb{E}_{\boldsymbol{c}, \pi^{\text{old}}(\boldsymbol{x}_0|\boldsymbol{c}), t} \, r \|\boldsymbol{v}_\theta^+(\boldsymbol{x}_t, \boldsymbol{c}, t) - \boldsymbol{v}\|_2^2 + (1 - r)\|\boldsymbol{v}_\theta^-(\boldsymbol{x}_t, \boldsymbol{c}, t) - \boldsymbol{v}\|_2^2, \qquad (10)$$

$$\text{where} \quad \boldsymbol{v}_\theta^+(\boldsymbol{x}_t, \boldsymbol{c}, t) := (1 - \beta)\boldsymbol{v}^{old}(\boldsymbol{x}_t, \boldsymbol{c}, t) + \beta \boldsymbol{v}_\theta(\boldsymbol{x}_t, \boldsymbol{c}, t), \quad \text{(Implicit positive policy)}$$

$$\text{and} \quad \boldsymbol{v}_\theta^-(\boldsymbol{x}_t, \boldsymbol{c}, t) := (1 + \beta)\boldsymbol{v}^{old}(\boldsymbol{x}_t, \boldsymbol{c}, t) - \beta \boldsymbol{v}_\theta(\boldsymbol{x}_t, \boldsymbol{c}, t). \quad \text{(Implicit negative policy)}$$

*Given unlimited data and model capacity, the optimal solution of Eq.* (10) *satisfies*

$$\boldsymbol{v}_{\theta^*}(\boldsymbol{x}_t, \boldsymbol{c}, t) = \boldsymbol{v}^{old}(\boldsymbol{x}_t, \boldsymbol{c}, t) + \frac{2}{\beta}\Delta(\boldsymbol{x}_t, \boldsymbol{c}, t).$$

*Proof.*

$$\mathcal{L}(\theta) = \mathbb{E}_{\boldsymbol{c}, t, \pi_t^{\text{old}}(\boldsymbol{x}_t|\boldsymbol{c})\pi_{0|t}^{\text{old}}(\boldsymbol{x}_0|\boldsymbol{x}_t, \boldsymbol{c})} \, r(\boldsymbol{x}_0, \boldsymbol{c})\|\boldsymbol{v}_\theta^+(\boldsymbol{x}_t, \boldsymbol{c}, t) - \boldsymbol{v}\|_2^2 + [1 - r(\boldsymbol{x}_0, \boldsymbol{c})]\|\boldsymbol{v}_\theta^-(\boldsymbol{x}_t, \boldsymbol{c}, t) - \boldsymbol{v}\|_2^2$$

$$= \mathbb{E}_{\boldsymbol{c}.t, \pi_t^{\text{old}}(\boldsymbol{x}_t|\boldsymbol{c})}\{\mathbb{E}_{\pi_{0|t}^{\text{old}}(\boldsymbol{x}_0|\boldsymbol{x}_t, \boldsymbol{c})} r(\boldsymbol{x}_0, \boldsymbol{c})\|\boldsymbol{v}_\theta^+(\boldsymbol{x}_t, \boldsymbol{c}, t) - \boldsymbol{v}\|_2^2$$

$$+ \mathbb{E}_{\pi_{0|t}^{\text{old}}(\boldsymbol{x}_0|\boldsymbol{x}_t, \boldsymbol{c})}[1 - r(\boldsymbol{x}_0, \boldsymbol{c})]\|\boldsymbol{v}_\theta^-(\boldsymbol{x}_t, \boldsymbol{c}, t) - \boldsymbol{v}\|_2^2\}$$

From Lemma A.1 we have $r(\boldsymbol{x}_0, \boldsymbol{c})\pi^{\text{old}}(\boldsymbol{x}_0|\boldsymbol{c}) = p_{\pi^{\text{old}}}(\mathbf{o} = 1|\boldsymbol{c})\pi^+(\boldsymbol{x}_0|\boldsymbol{c})$, therefore:

$$r(\boldsymbol{x}_0, \boldsymbol{c})\pi_{0|t}^{\text{old}}(\boldsymbol{x}_0|\boldsymbol{x}_t, \boldsymbol{c}) = r(\boldsymbol{x}_0, \boldsymbol{c})\frac{\pi^{\text{old}}(\boldsymbol{x}_0|\boldsymbol{c})\pi(\boldsymbol{x}_t|\boldsymbol{x}_0)}{\pi_t^{\text{old}}(\boldsymbol{x}_t|\boldsymbol{c})}$$

$$= p_{\pi^{\text{old}}}(\mathbf{o} = 1|\boldsymbol{c})\frac{\pi_t^+(\boldsymbol{x}_t|\boldsymbol{c})}{\pi_t^{\text{old}}(\boldsymbol{x}_t|\boldsymbol{c})}\frac{\pi^+(\boldsymbol{x}_0|\boldsymbol{c})\pi(\boldsymbol{x}_t|\boldsymbol{x}_0)}{\pi_t^+(\boldsymbol{x}_t|\boldsymbol{c})}$$

$$= p_{\pi^{\text{old}}}(\mathbf{o} = 1|\boldsymbol{c})\frac{\pi_t^+(\boldsymbol{x}_t|\boldsymbol{c})}{\pi_t^{\text{old}}(\boldsymbol{x}_t|\boldsymbol{c})}\pi_{0|t}^+(\boldsymbol{x}_0|\boldsymbol{x}_t, \boldsymbol{c})$$

$$= \alpha(\boldsymbol{x}_t)\pi_{0|t}^+(\boldsymbol{x}_0|\boldsymbol{x}_t, \boldsymbol{c})$$

Similarly,

$$[1 - r(\boldsymbol{x}_0, \boldsymbol{c})]\pi_{0|t}^{\text{old}}(\boldsymbol{x}_0|\boldsymbol{x}_t, \boldsymbol{c}) = [1 - \alpha(\boldsymbol{x}_t)]\pi_{0|t}^-(\boldsymbol{x}_0|\boldsymbol{x}_t, \boldsymbol{c})$$

Then,

$$\mathcal{L}(\theta) = \mathbb{E}_{\boldsymbol{c}, t, \pi_t^{\text{old}}(\boldsymbol{x}_t|\boldsymbol{c})}\{\alpha(\boldsymbol{x}_t)\mathbb{E}_{\pi_{0|t}^+(\boldsymbol{x}_0|\boldsymbol{x}_t, \boldsymbol{c})}\|\boldsymbol{v}_\theta^+(\boldsymbol{x}_t, \boldsymbol{c}, t) - \boldsymbol{v}\|_2^2$$

$$+ [1 - \alpha(\boldsymbol{x}_t)]\mathbb{E}_{\pi_{0|t}^-(\boldsymbol{x}_0|\boldsymbol{x}_t, \boldsymbol{c})}\|\boldsymbol{v}_\theta^-(\boldsymbol{x}_t, \boldsymbol{c}, t) - \boldsymbol{v}\|_2^2\}$$

$$= \mathbb{E}_{\boldsymbol{c}, t, \pi_t^{\text{old}}(\boldsymbol{x}_t|\boldsymbol{c})}\{\alpha(\boldsymbol{x}_t)\|\boldsymbol{v}_\theta^+(\boldsymbol{x}_t, \boldsymbol{c}, t) - \mathbb{E}_{\pi_{0|t}^+(\boldsymbol{x}_0|\boldsymbol{x}_t, \boldsymbol{c})}[\boldsymbol{v}]\|_2^2$$

$$+ [1 - \alpha(\boldsymbol{x}_t)]\|\boldsymbol{v}_\theta^-(\boldsymbol{x}_t, \boldsymbol{c}, t) - \mathbb{E}_{\pi_{0|t}^-(\boldsymbol{x}_0|\boldsymbol{x}_t, \boldsymbol{c})}[\boldsymbol{v}]\|_2^2\} + C_1$$

$$= \mathbb{E}_{\boldsymbol{c}, t, \pi_t^{\text{old}}(\boldsymbol{x}_t|\boldsymbol{c})}\{\alpha(\boldsymbol{x}_t)\|\boldsymbol{v}_\theta^+(\boldsymbol{x}_t, \boldsymbol{c}, t) - \boldsymbol{v}^+(\boldsymbol{x}_t, \boldsymbol{c}, t)\|_2^2$$

$$+ [1 - \alpha(\boldsymbol{x}_t)]\|\boldsymbol{v}_\theta^-(\boldsymbol{x}_t, \boldsymbol{c}, t) - \boldsymbol{v}^-(\boldsymbol{x}_t, \boldsymbol{c}, t)\|_2^2\} + C_1$$

Combining Theorem A.3, we observe that

$$\boldsymbol{v}_\theta^+(\boldsymbol{x}_t, \boldsymbol{c}, t) - \boldsymbol{v}^+(\boldsymbol{x}_t, \boldsymbol{c}, t) = (1 - \beta)\boldsymbol{v}^{\text{old}}(\boldsymbol{x}_t, \boldsymbol{c}, t) + \beta\boldsymbol{v}_\theta(\boldsymbol{x}_t, \boldsymbol{c}, t) - \boldsymbol{v}^+(\boldsymbol{x}_t, \boldsymbol{c}, t)$$

$$= \beta[\boldsymbol{v}_\theta - \boldsymbol{v}^{\text{old}} - \frac{1}{\beta}\frac{\Delta}{\alpha(\boldsymbol{x}_t)}]$$

$$\boldsymbol{v}_\theta^-(\boldsymbol{x}_t, \boldsymbol{c}, t) - \boldsymbol{v}^-(\boldsymbol{x}_t, \boldsymbol{c}, t) = (1 + \beta)\boldsymbol{v}^{\text{old}}(\boldsymbol{x}_t, \boldsymbol{c}, t) - \beta\boldsymbol{v}_\theta(\boldsymbol{x}_t, \boldsymbol{c}, t) - \boldsymbol{v}^-(\boldsymbol{x}_t, \boldsymbol{c}, t)$$

$$= -\beta[\boldsymbol{v}_\theta - \boldsymbol{v}^{\text{old}} - \frac{1}{\beta}\frac{\Delta}{1 - \alpha(\boldsymbol{x}_t)}]$$

Substituting these results into $\mathcal{L}(\theta)$:

$$\mathcal{L}(\theta) = \mathbb{E}_{\boldsymbol{c}, t, \pi_t^{\text{old}}(\boldsymbol{x}_t|\boldsymbol{c})}\{\alpha(\boldsymbol{x}_t)\beta^2\|\boldsymbol{v}_\theta - \boldsymbol{v}^{\text{old}} - \frac{1}{\beta}\frac{\Delta}{\alpha(\boldsymbol{x}_t)}\|_2^2$$

$$+ [1 - \alpha(\boldsymbol{x}_t)]\beta^2\|\boldsymbol{v}_\theta - \boldsymbol{v}^{\text{old}} - \frac{1}{\beta}\frac{\Delta}{1 - \alpha(\boldsymbol{x}_t)}\|_2^2\} + C_1$$

$$= \beta^2\mathbb{E}_{\boldsymbol{c}, t, \pi_t^{\text{old}}(\boldsymbol{x}_t|\boldsymbol{c})}\{\alpha(\boldsymbol{x}_t)\|\boldsymbol{v}_\theta - (\boldsymbol{v}^{\text{old}} + \frac{1}{\beta}\frac{\Delta}{\alpha(\boldsymbol{x}_t)})\|_2^2$$

$$+ [1 - \alpha(\boldsymbol{x}_t)]\|\boldsymbol{v}_\theta - (\boldsymbol{v}^{\text{old}} + \frac{1}{\beta}\frac{\Delta}{1 - \alpha(\boldsymbol{x}_t)})\|_2^2\} + C_1$$

$$= \beta^2\mathbb{E}_{\boldsymbol{c}, t, \pi^{\text{old}}(\boldsymbol{x}_t|\boldsymbol{c})}\|\boldsymbol{v}_\theta - \alpha(\boldsymbol{x}_t)(\boldsymbol{v}^{\text{old}} + \frac{1}{\beta}\frac{\Delta}{\alpha(\boldsymbol{x}_t)}) - [1 - \alpha(\boldsymbol{x}_t)](\boldsymbol{v}^{\text{old}} + \frac{1}{\beta}\frac{\Delta}{1 - \alpha(\boldsymbol{x}_t)})\|_2^2 + C_2$$

$$= \beta^2\mathbb{E}_{\boldsymbol{c}, t, \pi^{\text{old}}(\boldsymbol{x}_t|\boldsymbol{c})}\|\boldsymbol{v}_\theta - (\boldsymbol{v}^{\text{old}} + \frac{2}{\beta}\Delta)\|_2^2 + C_2$$

from which it is obvious that the optimal $\theta^*$ satisfies $\boldsymbol{v}_{\theta^*}(\boldsymbol{x}_t, \boldsymbol{c}, t) = \boldsymbol{v}^{\text{old}}(\boldsymbol{x}_t, \boldsymbol{c}, t) + \frac{2}{\beta}\Delta(\boldsymbol{x}_t, \boldsymbol{c}, t)$. $\square$

## B  THEORETICAL DISCUSSIONS

### B.1  FLOW SDE

As flow models are a special case of diffusion models under the rectified schedule $\alpha_t = 1 - t, \sigma_t = t$, the earliest results on diffusion SDEs (Song et al., 2020b) can be directly applied without difficulty. FlowGRPO (Liu et al., 2025) and DanceGRPO (Xue et al., 2025) derive the flow SDE with unexplained hyperparameters $g_t = a\sqrt{\frac{t}{1-t}}$ or additional complexity. We provide a simpler and more principled perspective based solely on the diffusion model framework.

To leverage the diffusion SDE formulation in Song et al. (2020b), we need to match its forward SDE $\mathrm{d}\boldsymbol{x}_t = f(t)\boldsymbol{x}_t\mathrm{d}t + g(t)\mathrm{d}\boldsymbol{w}_t$ with the forward transition kernel $\boldsymbol{x}_t = \alpha_t\boldsymbol{x}_0 + \sigma_t\boldsymbol{\epsilon}$. As noted in the first two arXiv versions of the VDM paper (Kingma et al., 2021), $f(t), g(t)$ are related to $\alpha_t, \sigma_t$ by $f(t) = \frac{\mathrm{d}\log\alpha_t}{\mathrm{d}t}, g^2(t) = \frac{\mathrm{d}\sigma_t^2}{\mathrm{d}t} - 2\frac{\mathrm{d}\log\alpha_t}{\mathrm{d}t}\sigma_t^2$. Setting $\alpha_t = 1 - t, \sigma_t = t$, we have

$$f(t) = -\frac{1}{1-t}, \quad g^2(t) = \frac{2t}{1-t} \tag{11}$$

for rectified flow. According to (Huang et al., 2021), the generalized reverse SDE takes the form:

$$\mathrm{d}\boldsymbol{x}_t = \left[f(t)\boldsymbol{x}_t - \frac{1 + \lambda_t^2}{2}g^2(t)\nabla_{\boldsymbol{x}_t}\log\pi_t(\boldsymbol{x}_t)\right]\mathrm{d}t + \lambda_t g(t)\mathrm{d}\bar{\boldsymbol{w}}_t \tag{12}$$

where $\lambda_t \in [0, 1]$. Equivalently, it amounts to introducing Langevin dynamics on top of the diffusion ODE, with $\lambda_t = 0$ corresponding to ODE, and $\lambda_t = 1$ corresponding to the maximum variance SDE in Song et al. (2020b). The score function $\boldsymbol{s}_\theta(\boldsymbol{x}_t, t) \approx \nabla_{\boldsymbol{x}_t}\log\pi_t(\boldsymbol{x}_t)$, noise predictor $\boldsymbol{\epsilon}_\theta(\boldsymbol{x}_t, t)$, data predictor $\boldsymbol{x}_\theta(\boldsymbol{x}_t, t)$ and velocity predictor $\boldsymbol{v}_\theta(\boldsymbol{x}_t, t)$ are interconvertible under general noise schedules (Zheng et al., 2023b):

$$\boldsymbol{\epsilon}_\theta(\boldsymbol{x}_t, t) = -\sigma_t\boldsymbol{s}_\theta(\boldsymbol{x}_t, t), \quad \boldsymbol{x}_\theta(\boldsymbol{x}_t, t) = \frac{\boldsymbol{x}_t - \sigma_t\boldsymbol{\epsilon}_\theta(\boldsymbol{x}_t, t)}{\alpha_t}, \quad \boldsymbol{v}_\theta(\boldsymbol{x}_t, t) = \dot{\alpha}_t\boldsymbol{x}_\theta(\boldsymbol{x}_t, t) + \dot{\sigma}_t\boldsymbol{\epsilon}_\theta(\boldsymbol{x}_t, t) \tag{13}$$

Applying these relations to the rectified flow schedule, we can derive:

$$s_\theta(\boldsymbol{x}_t, t) = -\frac{\boldsymbol{x}_t + (1-t)\boldsymbol{v}_\theta(\boldsymbol{x}_t, t)}{t} \tag{14}$$

Substituting Eq. (11) and Eq. (14) into Eq. (12), we have the diffusion SDE under rectified flow:

$$\mathrm{d}\boldsymbol{x}_t = \left[(1+\lambda_t^2)\boldsymbol{v}_\theta(\boldsymbol{x}_t, t) + \frac{\lambda_t^2}{1-t}\boldsymbol{x}_t\right]\mathrm{d}t + \lambda_t\sqrt{\frac{2t}{1-t}}\mathrm{d}\boldsymbol{w} \tag{15}$$

Therefore, the flow SDE in Eq. (2) is essentially conducting a transformation $g_t = \lambda_t\sqrt{\frac{2t}{1-t}}$ from the interpolation parameter $\lambda_t \in [0, 1]$ to the variance parameter $g_t$. This also explains the choice $g_t = a\sqrt{\frac{t}{1-t}}$ in FlowGRPO, where $a = \sqrt{2}\lambda_t$ is a scaled version of $\lambda_t$, with $a = \sqrt{2}$ corresponding to the maximum variance SDE. In comparison, DanceGRPO adopts a fixed variance $g_t$ across timesteps, which is less effective on image models while more stable on video models.

FlowGRPO and DanceGRPO directly take the Euler discretization of the flow SDE. In principle, there are more accurate ways, such as utilizing the idea of diffusion implicit models (Song et al., 2020a; Zheng et al., 2024), which is equivalent to the first-order discretization after applying exponential integrators (Hochbruck & Ostermann, 2010; Zhang & Chen, 2022; Gonzalez et al., 2023). Specifically, the sampling step from $t$ to $s < t$ can be derived as:

$$\boldsymbol{x}_s = \left[(1-s) + \sqrt{s^2 - \rho_t^2}\right]\boldsymbol{x}_t - \left[(1-s)t - \sqrt{s^2 - \rho_t^2}(1-t)\right]\boldsymbol{v}_\theta(\boldsymbol{x}_t, t) + \rho_t\boldsymbol{\epsilon}, \quad \boldsymbol{\epsilon} \sim \mathcal{N}(\mathbf{0}, \mathbf{I}) \tag{16}$$

where $\rho_t = \eta_t s\sqrt{1 - \frac{s^2(1-t)^2}{t^2(1-s)^2}}$, and $\eta_t \in [0, 1]$ interpolates between ODE and maximum variance SDE. Compared to the Euler discretization, the DDIM-style discretization avoids singularities at boundaries and is expected to reduce sampling errors. However, we did not observe notable advantages by replacing the SDE sampler with stochastic DDIM. Concurrent work (Wang & Yu, 2025) improves the SDE sampler through the Coefficients-Preserving Sampling (CPS) principle.

## B.2 HIGH-ORDER FLOW ODE SAMPLER

We implement the 2nd-order ODE sampler for flow models based on the DPM-Solver series (Lu et al., 2022a;b; Zheng et al., 2023a), which uses the multistep method and half the log signal-to-noise ratio (SNR) $\lambda_t = \log(\alpha_t/\sigma_t)$ for time discretization. Specifically, for three consecutive timesteps $t_i < t_{i-1} < t_{i-2}$, where $\boldsymbol{x}_{t_{i-1}}, \boldsymbol{x}_{t_{i-2}}$ are already obtained, the update rule for $\boldsymbol{x}_{t_i}$ is:

$$\boldsymbol{x}_{t_i} = \frac{\sigma_{t_i}}{\sigma_{t_{i-1}}}\boldsymbol{x}_{t_{i-1}} - \alpha_{t_i}(e^{-h_i} - 1)\left[\left(1 + \frac{1}{2r_i}\right)\boldsymbol{x}_\theta(\boldsymbol{x}_{t_{i-1}}, t_{i-1}) - \frac{1}{2r_i}\boldsymbol{x}_\theta(\boldsymbol{x}_{t_{i-2}}, t_{i-2})\right] \tag{17}$$

where $h_i = \lambda_{t_i} - \lambda_{t_{i-1}}, r_i = \frac{h_{i-1}}{h_i}$, and the data predictor $\boldsymbol{x}_\theta = \boldsymbol{x}_t - t\boldsymbol{v}_\theta$ for rectified flow. High-order solvers are also adopted in MixGRPO (Li et al., 2025b) but only for certain steps. Adopting the 2nd-order solver throughout the entire sampling process is infeasible, as $\lambda_t$ will be infinity at boundaries $t = 0$ or $t = 1$. Following common practices, the first and last steps degrade to the first-order solver, which is the default Euler discretization for flow models.

## B.3 INTUITION BEHIND THE FLOWGRPO OBJECTIVE

We provide some insight into reverse-process diffusion RL by inspecting the FlowGRPO objective in a sampler-agnostic manner. For any first-order SDE sampler, the reverse sampling step from $t$ to $s < t$ can be expressed as

$$\boldsymbol{x}_s = l(s, t)\boldsymbol{x}_t - m(s, t)\boldsymbol{v}_\theta(\boldsymbol{x}_t, t) + n(s, t)\boldsymbol{\epsilon}, \quad \boldsymbol{\epsilon} \sim \mathcal{N}(\mathbf{0}, \mathbf{I}) \tag{18}$$

where $l(s, t), m(s, t), n(s, t)$ depend only on $s, t$ and the sampler. Consider the on-policy case and the branching strategy in MixGRPO. Starting from a shared $\boldsymbol{x}_t$, a group of $N$ noises $\boldsymbol{\epsilon}^{(1)}, \ldots, \boldsymbol{\epsilon}^{(N)}$ are sampled and incorporated into the reverse step to produce multiple samples $\boldsymbol{x}_s^{(1)}, \ldots, \boldsymbol{x}_s^{(N)}$.

They go through further sampling, yielding $N$ clean samples and corresponding advantages $A^{(1)}, \ldots, A^{(N)}$. On-policy GRPO minimizes the negative advantage-weighted log likelihoods:

$$\mathcal{L}(\theta) = -\frac{1}{N} \sum_{i=1}^{N} A^{(i)} \log p_\theta(\boldsymbol{x}_s^{(i)} | \boldsymbol{x}_t) \tag{19}$$

where

$$\begin{aligned}
\log p_\theta(\boldsymbol{x}_s^{(i)} | \boldsymbol{x}_t) &= -\frac{\|\boldsymbol{x}_s^{(i)} - (l(s,t)\boldsymbol{x}_t - m(s,t)\boldsymbol{v}_\theta(\boldsymbol{x}_t, t))\|_2^2}{2n^2(s,t)} + C \\
&= -\frac{\|m(s,t)\boldsymbol{v}_\theta(\boldsymbol{x}_t, t) - m(s,t)\boldsymbol{v}_{\mathsf{sg}(\theta)}(\boldsymbol{x}_t, t) + n(s,t)\boldsymbol{\epsilon}^{(i)}\|_2^2}{2n^2(s,t)} + C
\end{aligned} \tag{20}$$

$\mathsf{sg}$ emerges because the samples $\boldsymbol{x}_s^{(1)}, \ldots, \boldsymbol{x}_s^{(N)}$ are gradient-free. The gradient of the reverse-step log likelihood w.r.t. $\theta$ can be surprisingly reduced to a simple form:

$$\nabla_\theta \log p_\theta(\boldsymbol{x}_s^{(i)} | \boldsymbol{x}_t) = -\frac{m(s,t)}{n(s,t)} \nabla_\theta((\boldsymbol{\epsilon}^{(i)})^\top \boldsymbol{v}_\theta(\boldsymbol{x}_t, t)) \tag{21}$$

and

$$\nabla_\theta \mathcal{L}(\theta) = \frac{m(s,t)}{n(s,t)} \nabla_\theta \left[ \frac{1}{N} \sum_{i=1}^{N} (A^{(i)} \boldsymbol{\epsilon}^{(i)})^\top \boldsymbol{v}_\theta(\boldsymbol{x}_t, t) \right] \tag{22}$$

Therefore, FlowGRPO essentially aligns the velocity field with the *advantage-weighted noise*, while the choice of timesteps and sampler only influences the weighting $\frac{m(s,t)}{n(s,t)}$ across sampling steps. In the following, we show a further conclusion that FlowGRPO can be viewed as *a gradient estimation of reward backpropagation*.

Denote $r_t(\boldsymbol{x}_t)$ as the implicit gradient-free function that solves the PF-ODE from $t$ to $0$ and fetches the reward on the cleaned sample. The rewards can be expressed as

$$r^{(i)} = r_s \left( l(s,t)\boldsymbol{x}_t - m(s,t)\boldsymbol{v}_\theta(\boldsymbol{x}_t, t) + n(s,t)\boldsymbol{\epsilon}^{(i)} \right) \tag{23}$$

According to Stein's identity, we have

$$\begin{aligned}
\frac{1}{N} \sum_{i=1}^{N} r^{(i)} \boldsymbol{\epsilon}^{(i)} &\approx \mathbb{E}_{\boldsymbol{\epsilon} \sim \mathcal{N}(\boldsymbol{0}, \boldsymbol{I})} \left[ r_s \left( l(s,t)\boldsymbol{x}_t - m(s,t)\boldsymbol{v}_\theta(\boldsymbol{x}_t, t) + n(s,t)\boldsymbol{\epsilon} \right) \boldsymbol{\epsilon} \right] \\
&= n(s,t) \mathbb{E}_{\boldsymbol{\epsilon} \sim \mathcal{N}(\boldsymbol{0}, \boldsymbol{I})} \left[ \nabla r_s \left( l(s,t)\boldsymbol{x}_t - m(s,t)\boldsymbol{v}_\theta(\boldsymbol{x}_t, t) + n(s,t)\boldsymbol{\epsilon} \right) \right]
\end{aligned} \tag{24}$$

Therefore,

$$\begin{aligned}
&\nabla_\theta \left[ \frac{1}{N} \sum_{i=1}^{N} (A^{(i)} \boldsymbol{\epsilon}^{(i)})^\top \boldsymbol{v}_\theta(\boldsymbol{x}_t, t) \right] \\
&\approx \frac{n(s,t)}{\sigma} \mathbb{E}_{\boldsymbol{\epsilon} \sim \mathcal{N}(\boldsymbol{0}, \boldsymbol{I})} \left[ \nabla r_s \left( l(s,t)\boldsymbol{x}_t - m(s,t)\boldsymbol{v}_\theta(\boldsymbol{x}_t, t) + n(s,t)\boldsymbol{\epsilon} \right) \nabla_\theta \boldsymbol{v}_\theta(\boldsymbol{x}_t, t) \right] \\
&= -\frac{n(s,t)}{m(s,t)\sigma} \mathbb{E}_{\boldsymbol{\epsilon} \sim \mathcal{N}(\boldsymbol{0}, \boldsymbol{I})} \left[ \nabla_\theta r_s \left( l(s,t)\boldsymbol{x}_t - m(s,t)\boldsymbol{v}_\theta(\boldsymbol{x}_t, t) + n(s,t)\boldsymbol{\epsilon} \right) \right]
\end{aligned} \tag{25}$$

where $\sigma$ is the global std used in GRPO normalization. Therefore, the GRPO loss gradient is

$$\nabla_\theta \mathcal{L}(\theta) \approx -\frac{1}{\sigma} \mathbb{E}_{\boldsymbol{\epsilon} \sim \mathcal{N}(\boldsymbol{0}, \boldsymbol{I})} \left[ \nabla_\theta r_s \left( l(s,t)\boldsymbol{x}_t - m(s,t)\boldsymbol{v}_\theta(\boldsymbol{x}_t, t) + n(s,t)\boldsymbol{\epsilon} \right) \right] \tag{26}$$

From the above gradient, GRPO optimizes the reverse transition $t \to s$ when the remaining trajectory $s \to 0$ is gradient-free. Compared to works like ReFL (Xu et al., 2023), which conduct direct gradient backpropagation and approximate $s \to 0$ with a single forward pass ($\boldsymbol{x}_0$-prediction), GRPO introduces higher estimation variance but avoids backpropagation through the $s \to 0$ process, allowing larger $s$ and a longer sampling chain for $s \to 0$.

## C   EXPERIMENT DETAILS

**Training Configurations.** Our setup largely follows FlowGRPO, adopting the same number of groups per epoch (48), group size (24), LoRA configuration ($\alpha = 64, r = 32$), and learning rate ($3e - 4$). For each collected clean image, forward noising and loss computation are performed exactly on the corresponding sampling timesteps. We employ a 2nd-order ODE sampler for data collection and enable adaptive time weighting by default.

**Single-Reward.** For a head-to-head comparison with FlowGRPO under single-reward settings, we fix the number of sampling steps to 10 to ensure fairness. By default, we set $\beta = 1$ and $\eta_i = \min(0.001i, 0.5)$, which work stably for most reward models. In the case of OCR, the reward rapidly approaches 1 within 100 iterations but suffers from instability. To address this, we adopt a more conservative soft-update strategy with $\eta_{\max} = 0.999$.

**Multi-Reward.** To comprehensively improve the base model across multiple rewards, we adopt a multi-stage training scheme. The training setup involves three categories of rewards and datasets: (1) PickScore, CLIPScore, and HPSv2.1 rewards on the Pick-a-Pic dataset; (2) GenEval reward with the three rewards above on the GenEval dataset; and (3) OCR reward with the three rewards above on the OCR dataset. Since the initial CFG-free generation is of low quality, we first train on (1) for 800 iterations to enhance image quality, followed by (2) for 300 iterations, (1) for 200 iterations, (2) for 200 iterations, and finally (3) for 100 iterations. All rewards are equally weighted, with PickScore divided by 26 for normalization to $[0, 1]$. By default, we use $\beta = 0.1$ and $\eta_i = \min(0.001i, 0.5)$, while setting $\eta_{\max} = 0.95$ for OCR to stabilize training. The number of sampling steps is fixed to 40 to ensure high-fidelity data collection.

## D   ADDITIONAL RESULTS

Table 2: Evaluation results of FlowGRPO and DiffusionNFT trained on single rewards, both initialized from CFG-free base model. Gray-colored: In-domain reward. We observe that training exclusively on the OCR reward impairs generalization to other metrics; to compensate this, we enable CFG when evaluating non-OCR rewards for OCR-trained models.

| Model | #Iter | Rule-Based | | Model-Based | | | | | |
|---|---|---|---|---|---|---|---|---|---|
| | | **GenEval** | **OCR** | **PickScore** | **ClipScore** | **HPSv2.1** | **Aesthetic** | **ImgRwd** | **UniRwd** |
| SD3.5-M (w/o CFG) | — | 0.24 | 0.12 | 20.51 | 0.237 | 0.204 | 5.13 | -0.58 | 2.02 |
| + CFG | — | 0.63 | 0.59 | 22.34 | 0.285 | 0.279 | 5.36 | 0.85 | 3.03 |
| + FlowGRPO | 4k | 0.97 | 0.30 | 21.78 | 0.277 | 0.248 | 5.15 | 0.74 | 2.87 |
| | 1k | 0.66 | 0.96 | 21.94 | 0.280 | 0.257 | 5.18 | 0.31 | 2.86 |
| | 4k | 0.54 | 0.60 | 23.62 | 0.257 | 0.295 | 6.42 | 1.17 | 3.17 |
| + Ours | 1k | 0.98 | 0.36 | 21.92 | 0.271 | 0.251 | 5.33 | 0.68 | 2.91 |
| | 150 | 0.54 | 0.97 | 21.63 | 0.281 | 0.246 | 5.19 | 0.37 | 2.81 |
| | 2k | 0.53 | 0.64 | 24.03 | 0.270 | 0.315 | 6.17 | 1.29 | 3.40 |

We provide more qualitative comparison between the base model, FlowGRPO and our multi-reward optimized model in Figure 11, Figure 12 and Figure 13.

| SD3.5-M (w/o CFG) | SD3.5-M (w/ CFG) | +FlowGRPO (w/ CFG) | +DiffusionNFT (w/o CFG) |
|---|---|---|---|

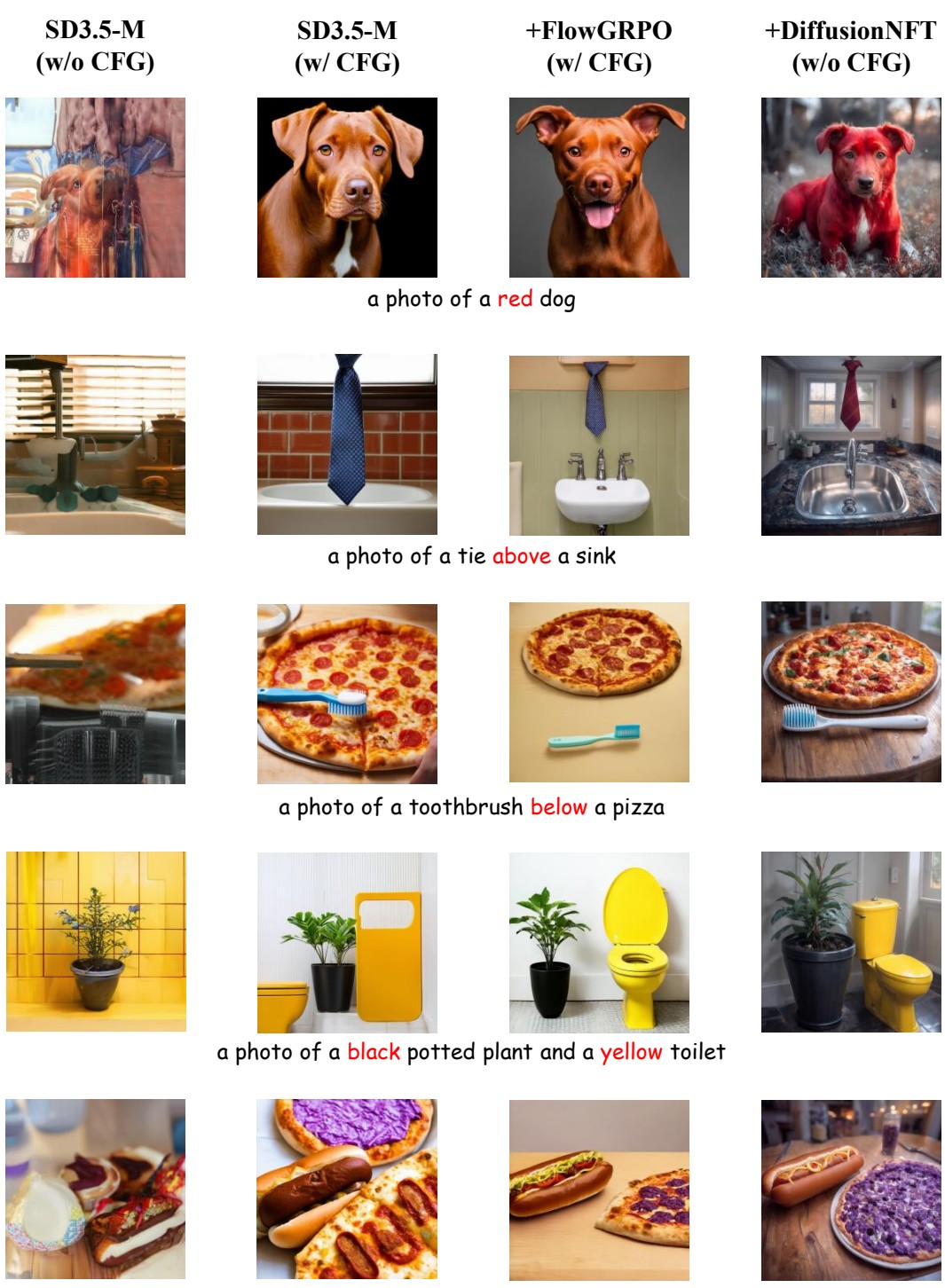

Figure 11: Qualitative comparison between FlowGRPO and our model on GenEval prompts.

| **SD3.5-M (w/o CFG)** | **SD3.5-M (w/ CFG)** | **+FlowGRPO (w/ CFG)** | **+DiffusionNFT (w/o CFG)** |
|---|---|---|---|
| 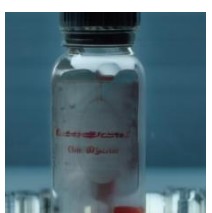 | 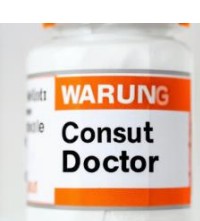 | 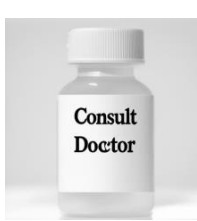 | 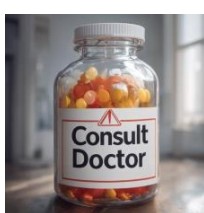 |

A close-up of a medicine bottle with a prominent warning label that reads "Consult Doctor", set against a neutral background, emphasizing the clarity and visibility of the text.

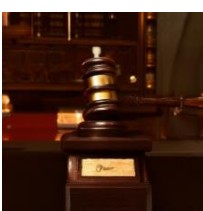 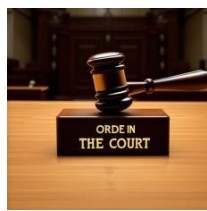 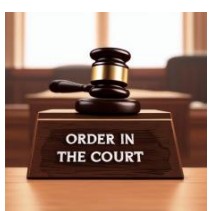 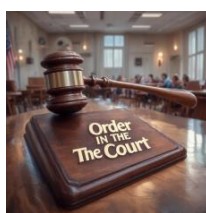

A courtroom scene with a judge's gavel resting on a wooden plaque that reads "Order in the Court", set against the backdrop of a quiet, solemn courtroom.

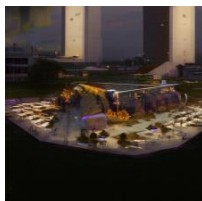 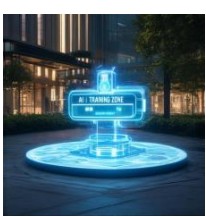 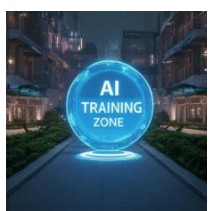 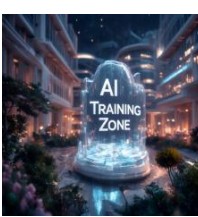

A realistic photo of a tech campus courtyard at night, featuring a glowing "AI Training Zone" hologram floating in the center, surrounded by futuristic buildings and greenery, with soft ambient lighting enhancing the futuristic atmosphere.

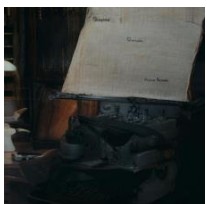 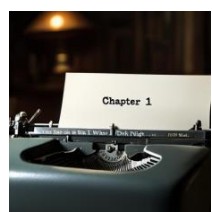 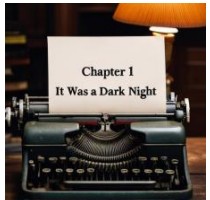 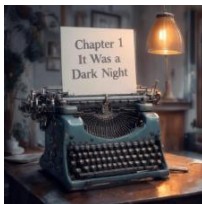

An antique typewriter with a sheet of paper inserted, prominently displaying the typed words: "Chapter 1 It Was a Dark Night". The scene is set in a dimly lit, vintage study with a single desk lamp casting a warm glow over the typewriter.

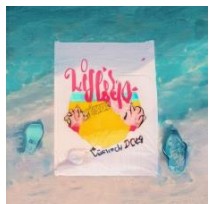 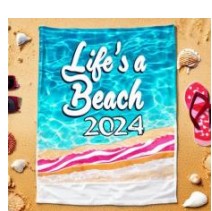 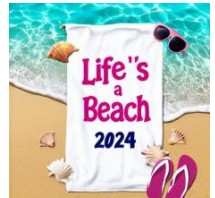 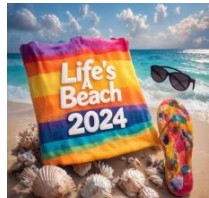

A vibrant beach scene featuring a colorful towel with the phrase "Life's a Beach 2024" prominently displayed, surrounded by seashells, sunglasses, and a flip-flop, set against a backdrop of clear blue water and golden sand.

Figure 12: Qualitative comparison between FlowGRPO and our model on OCR prompts.

| SD3.5-M
(w/o CFG) | SD3.5-M
(w/ CFG) | +FlowGRPO
(w/ CFG) | +DiffusionNFT
(w/o CFG) |
|---|---|---|---|

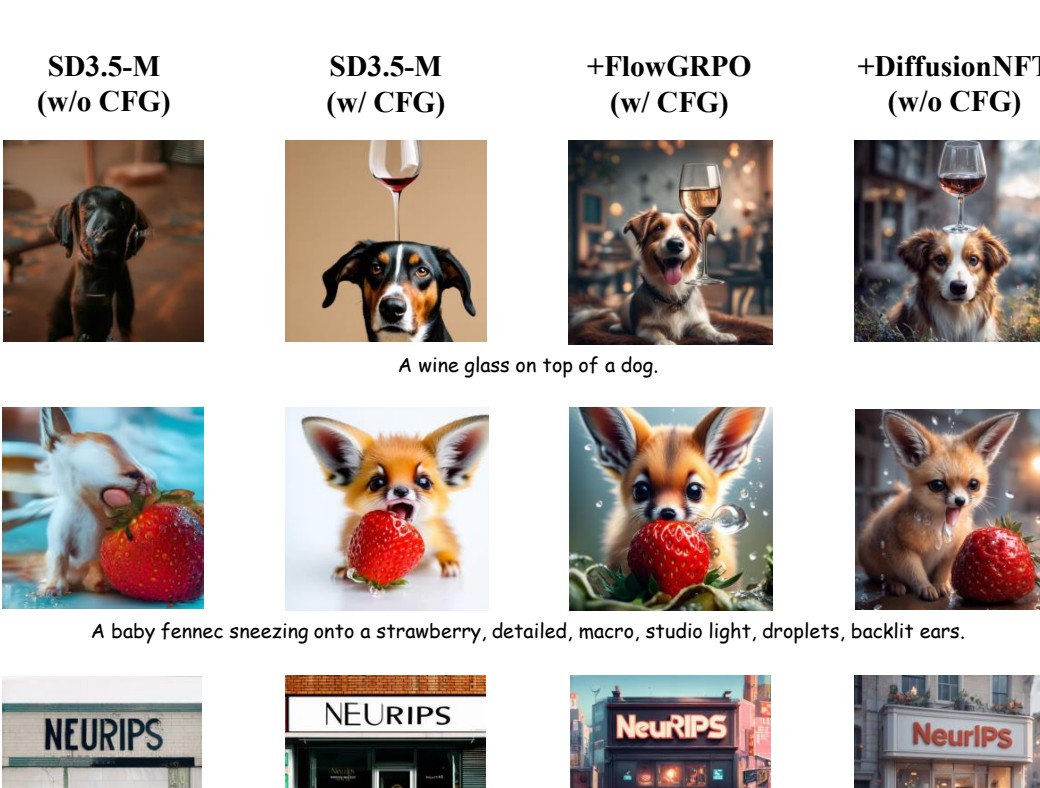

A wine glass on top of a dog.

A baby fennec sneezing onto a strawberry, detailed, macro, studio light, droplets, backlit ears.

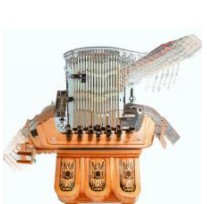 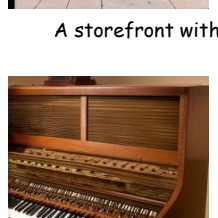 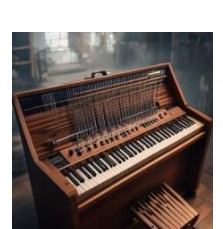 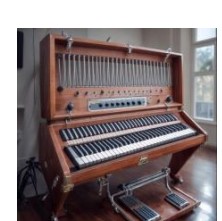

A storefront with 'NeurIPS' written on it.

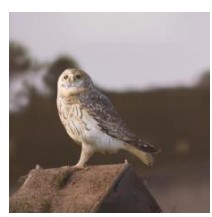 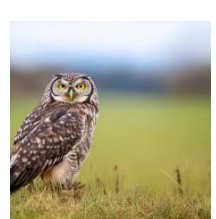 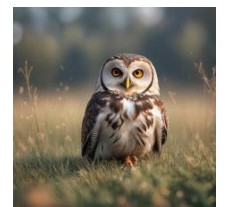 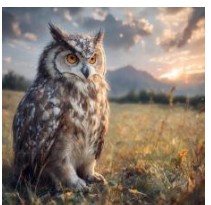

A large keyboard musical instrument with a wooden case enclosing a soundboard and metal strings, which are struck by hammers when the keys are depressed. The strings' vibration is stopped by dampers when the keys are released and can be regulated for length and volume by two or three pedals.

A side view of an owl sitting in a field.

Figure 13: Qualitative comparison between FlowGRPO and our model on DrawBench prompts.

