# OpenReview forum: "DiffusionNFT: Online Diffusion Reinforcement with Forward Process"
_ICLR.cc/2026/Conference — ICLR 2026 Oral_

### Official Review · Reviewer_w5yg · 2025-11-01

**Soundness:** 3
**Presentation:** 2
**Contribution:** 3
**Rating:** 8
**Confidence:** 3

**Summary:**

The paper introduces an online RL framework for optimizing diffusion models via flow matching on the forward process of the diffusion model.

**Strengths:**

The paper presents what is, to my knowledge, a novel approach to optimizing diffusion models via the forward process. The theoretical results seem sound, and DiffusionNFT shows drastic improvement in runtime efficiency over FlowGRPO. The approach can work with any solver for sampling, offering some generality/flexibility.

**Weaknesses:**

The paper compares only to FlowGRPO as the fine-tuning baseline. Other baselines such as [1] are not considered. Additionally, in Section 3.3, the paper discussed CFG-free optimization. How CFG is integrated as an offline form of RL guidance is slightly unclear and could benefit from greater exposition. Is $v^{old}$ updated during this offline RL style procedure?

Why does DiffusionNFT seem to dominate the Model-Based scores but not Rule-Based?

[1] Xue et al. “DanceGRPO: Unleashing GRPO on Visual Generation”

**Questions:**

See weaknesses section.

---

> ### Author Response · Authors · 2025-11-20
>
> We sincerely thank the reviewer for positive assessment of our work and insightful comments. We hope our response below can address the reviewer's concerns.
>
> > Limited baselines (DanceGRPO)
>
> We thank the reviewer for pointing out the highly relevant work. DanceGRPO and FlowGRPO are largely concurrent works that build upon the same fundamental reverse-process RL framework (GRPO). DanceGRPO is only slightly different to FlowGRPO in algorithm (same initial noise, different SDE variances), and its core RL paradigm, which we contrast with our forward-process approach, is the same as FlowGRPO's. Given that FlowGRPO has been optimized for rule-based rewards (GenEval, OCR), while DanceGRPO does not have such settings, we believe comparing to FlowGRPO on these rewards is sufficient evidence.
>
> > CFG integration and $v^{old}$ update in offline RL
>
> Our intention was to draw an analogy with CFG to build intuition, not to describe a procedure within our method. Standard CFG operates at inference time. It uses two separate, pre-trained, and fixed models (a conditional one and an unconditional one) to create a guidance direction. This can be considered as a form of "improvement from guidance" by utilizing a good (conditional) model and a bad (unconditional) model. We term this "offline" because the components are static and not learned during an online loop.
>
> DiffusionNFT operates during online training. It learns a similar guidance effect, but aims at reward improving and integrates it directly into the weights of a single model. At inference time, the DiffusionNFT optimized model could also incorporate CFG in an offline manner.
>
> The reviewer asks if $v^{old}$ is updated during this procedure. In the conceptual "offline" standard CFG, there is no update to any model. DiffusionNFT is developed under online settings. If someone wants to employ the DiffusionNFT loss to offline data, then $v^{old}$, the data collection policy, should not be updated.
>
> > Performance on Model-Based vs. Rule-Based Rewards
>
> The relatively small improvement on rule-based rewards compared to FlowGRPO is because GenEval/OCR measures accuracy and has a full score of 1.0. The performance of FlowGRPO (0.95/0.92) has been close to full score, and there is relatively little room for improvement. As generative models have stochasticity, we find it almost impossible to reach the perfect score (e.g., 0.98->1.00 on GenEval).

---

### Official Review · Reviewer_niMD · 2025-11-01

**Soundness:** 3
**Presentation:** 3
**Contribution:** 3
**Rating:** 6
**Confidence:** 4

**Summary:**

This paper introduces DiffusionNFT, a new RL paradigm that optimizes diffusion models directly on the forward process w.r.t. a reward function. It contrasts the positive and negative generations to define an implicit policy improvement direction. DiffusionNFT is compatible with any black-box ODE solvers and does not need to estimate the likelihood. It demonstrates significant improvement on SD3.5.

**Strengths:**

1. this paper reduces RL-based fine-tuning to supervised learning, which could be more stable and efficient compared to conventional Policy Gradient methods;
2. the training process utilizes both positive and negative signals
3. the empirical results demonstrate significant improvements

**Weaknesses:**

This paper optimizes the diffusion model on the forward process. While it brings several benefits, the distribution shift issue is not discussed in this paper. The training objective function is an expectation over the whole dataset. But the inference procedure only covers the region of trajectories from the positive samples. As a result, it's not very clear if the negative samples are used efficiently during training.

**Questions:**

See weakness.

---

> ### Author Response · Authors · 2025-11-20
>
> We sincerely thank the reviewer for positive assessment of our work and insightful comments. We hope our response below can address the reviewer's concerns.
>
> > Distribution shift and efficient use of negative samples
>
> This is an excellent and important question. We would like to clarify this point from two perspectives: 1) how negative samples are efficiently used, and 2) how our algorithm design explicitly handles the distribution shift inherent in online RL.
>
> 1. The essential role of negative samples in policy improvement
>
> The reviewer's concern that "inference procedure only covers the region of trajectories from the positive samples" is a potential misunderstanding we would like to clarify. Our training does not simply focus on positive samples. Instead, we learn a policy improvement direction, $\Delta$, which shifts the learned distribution $\pi_\theta$ towards generating higher-reward samples. The negative samples, which receive low rewards ($r\rightarrow 0$), are not only used to define the improved direction, but also serve as training examples of the negative policy ($v_\theta^-$, which is implicitly defined by $v_\theta,v_{old}$) in Equation 5. They teach the model which directions in the latent space to avoid.
>
> Empirically, we state in Section 4.4 that "Without the negative policy loss on $v_\theta^-$, we find rewards collapse almost instantly during online training, highlighting the essential role of negative signals in diffusion RL." This experimental result confirms that negative samples are not only used efficiently, but also essential for our method's success and stability.
>
> 2. Handling distribution shift in online RL
>
> The distribution shift is indeed a critical challenge in online RL. As our training policy $\pi_\theta$ improves, the data it generates comes from a new, shifted distribution. We have carefully considered this and incorporated a standard and effective mechanism to mitigate it. As described in Section 3.3, we employ a "soft" Exponential Moving Average (EMA) update for the data collection policy. This is a deliberate design choice to manage the trade-off between learning speed and stability. A strictly on-policy update would discard old data immediately. As we show in our ablation study in Figure 8, this leads to "rapid initial progress but is prone to severe instability.". Our soft update scheme ensures that the data collection policy does not stray too far or too quickly from the base policy, which mitigates the effects of distribution shift and stabilizes the entire online learning loop.

---

### Official Review · Reviewer_fF9P · 2025-11-02

**Soundness:** 4
**Presentation:** 2
**Contribution:** 3
**Rating:** 8
**Confidence:** 4

**Summary:**

This paper proposes to optimize diffusion models directly through forward process, opposed to prior works that will discretize the reverse process. The proposed algorithm yields a supervised learning objective for diffusion models fine-tuning as an off-policy RL paradigm, achieveing comparable performance to mainstream online RL based methods while using much less compute.

**Strengths:**

1. The general structure of this paper is clear. The introductions and preliminaries are clean and instinct. The paper explains clearly about the motivations, existing methods' limitations and the advantages of proposed methods.

2. There are several insightful theorems proposed that characterizes the policy improvement direction and a tractable supervised learning objective for policy optimization.

3. The experimental results are quite impressive in terms of achieveing comparable performance to the RL based counterparts like Flow-GRPO while using much less compute.

**Weaknesses:**

1. There lacks important baselines for comparisons, e.g. iterative DPO based methods for diffusion models, which also proposed supervised learning based objectives sampled from on-policy data. Overall the Diffusion-NFT objectives are quite similar, more like some weighted version of iterative Diffusion KTO, thus it will be beneficial to discuss the contributions of this paper over these baseline methods including the direct comparison of performance.

2. There lacks more recent models performance like FLUX and Video Generative Models, the reported results are only limited to Stable Diffusion Medium. Given the claimed efficiency of the proposed method, it seems practical to include the results of these other models.

3. Some presentations and clarity of the paper still needs to be improved. There are many missing explanations of the equation elements, for example:
* $o$ in Line 156 and $p_{\pi_{\text{old}}}$ in Line 161 are not explained. In addition, why introducing the two concepts of positive dataset and negative data in the online RL part, which seems irrelevant?

**Questions:**

1. The definition of reward functions in Line 156 is non standard. Why adopting such kind of reward and its optimality probability, opposed to the standard human satisfactory about the generations as an elo rating in existing works like DDPO and FlowGRPO? Specifically most existing reward models are also trained based on elo ratings, the current setup is a bit confusing. In addition to the pratical transformation in Section 3.3, ss there any theorectical equivalence or connection of the optimal solutions with the standard setup?

2. Why the forward consistency is ensured? There is no direct proof on this part. Especially I do not see the straightfoward difference between Diffusion-NFT and RL based method in ensuring such, since $x_t$ sampled from adding noise to $x_0$ is not from the true trajectory that samples $x_0$ in Diffusion-NFT. It will be beneficial to see experiments to showcase the possible distriution shift of RL based methods compared to Diffusion-NFT.

---

> ### Author Response · Authors · 2025-11-20
>
> We sincerely thank the reviewer for the detailed and constructive feedback. We hope our response below can address the reviewer's concerns.
>
> > Lacks important baselines (e.g., iterative DPO/KTO)
>
> We thank the reviewer for raising this important point, especially Diffusion-KTO which we did not discuss in the submission. When used in online setting, both NFT and KTO/DPO utilize positive/negative signals and the diffusion MSE loss terms, so they do appear similar. We would like to discuss the key differences below and revise the paper to include these discussions.
>
> For Diffusion-DPO, in fact, FlowGRPO has experimented with its online version (Appendix C.1 in their paper) and found it performs worse, especially on rule-based GenEval. We believe the deficiency of Diffusion-DPO is obvious because it is likelihood-based, i.e., the diffusion MSE is used to estimate log-likelihood through ELBO. As a result, it relies on Jensen's inequality to simplify the loss, which could introduce systematic bias. Moreover, it is constrained in a more narrow setting of paired preference optimization. Overall, Diffusion-DPO is **unprincipled** in some aspects, which could explain its inferior performance to FlowGRPO, even in online setting.
>
> For Diffusion-KTO, if ignoring its utility function $U$ and reference point $Q_{ref}$, its loss in v-pred form will be like
>
> $$
> E_{\pi_{ref}(x_0)}[w(x_0)(\|v_\theta(x_t,t)-v\|_2^2-\|v_{ref}(x_t,t)-v\|_2^2)]
> $$
>
> where $w(x_0)=\pm 1$ for positive/negative $x_0$. The second term has no impact on the gradient, even we replace $\pi_{ref}$ with $\pi_{old}$ in an online setting. Therefore, the loss is essentially diffusion loss on positive samples minus diffusion loss on negative samples.
>
> In the binary reward setting, the above loss will be exactly **a special case of *on-policy* DiffusionNFT under $\beta=1$**, i.e., $\pi_{old}=\pi_{sg(\theta)}$, and $v_\theta^+=v_\theta,v_\theta^-=2v_{sg(\theta)}-v_\theta$. We omit the proof here. With utility function $U$ and reference point $Q_{ref}$ incorporated, you are **correct and insightful** that DiffusionNFT in this special case is equivalent to some weighted version of iterative Diffusion KTO.
>
> Compared to iterative Diffusion KTO, the most critical discrepancy is that DiffusionNFT effectively incorporates $v_{old}$ into the loss as a reference point for improvement in **off-policy** settings ($\pi_{old}\neq\pi_{\theta}$). Off-policy training is important for stability, as observed in our ablation studies, and plays a significant role in large-scale RL. We believe the most important contribution of DiffusionNFT is that it provides a **principled way of off-policy diffusion reinforcement** without estimating intractable likelihood ratios.
>
> > Lacks more recent models (e.g., FLUX)
>
> We agree. As a work that focuses on developing new methods, we align our setting with FlowGRPO to ensure a rigorous, fair comparison and facilitate the exploration of design choices. We additionally apply DiffusionNFT to FLUX on the GenEval reward, the result is as follows:
>
> |Iterations|200|1000|
> |:--------|:-------:|:-------:|
> |FlowGRPO|0.68|0.91|
> |DiffusionNFT|0.93|0.96|
>
> As FLUX is similar to the SD3.5-Medium text-to-image setting, we believe DiffusionNFT's overall performance will also be similar. Applying it to video diffusion models like Wan will be more challenging, and we leave the exploration for future work.
>
> > Clarity of presentation, notations, and reward definition
>
> We apologize for the lack of clarity and missing explanations. We appreciate the reviewer pointing these out and will revise them.
>
> - Explanation of $o$, $p_{old}(o=1|c)$ and positive/negative datasets in online RL
>
> The concept of positive/negative datasets $D^+,D^-$ is the core of online RL discussed in our paper. As long as we learn a new policy that generates $D^+$, we obtain an improvement.
>
> $o$ is a binary random variable used to split the data strictly into $D^+$ and $D^-$, which is the foundation of our following definitions and derivations. $o=1$ means that a generated sample $x_0$ is "optimal" or "positive," and $o=0$ means it is "non-optimal" or "negative." This is a standard formulation for framing RL as a probabilistic inference problem, as introduced in Levine (2018), which we cite, and widely used, for example in rule-based math/code rewards of LLMs.
>
> $p_{old}(o=1|c)$ is for adapting to the continuous reward setting. For normalized reward $r(x_0, c)\in[0,1]$, it may not strictly be 0 or 1, so we frame it as **the probability to be allocated into the positive dataset** ($o=1$).
>
> - Non-standard reward $r(x_0, c) := p(o=1|x_0, c)$ vs. Elo:
>
> As we explained above, these transformations are used simply to adapt continuous, unbounded rewards, to our definition of strictly splitted positive/negative datasets. The raw reward score $r_{raw}$ can be from any reward model, including those based on Elo, PickScore, etc.

---

> > ### Author Response · Authors · 2025-11-20
> >
> > > Understanding of forward consistency
> >
> > Forward consistency is mainly a theoretical discussion. It can be intuitively understood as follows: for a valid diffusion model, its ground-truth noised marginal distributions $\pi_t(x_t)$ for any $t>0$ are determined once given the clean marginal distribution $\pi_0(x_0)$. The relationship comes from the pre-defined forward process. More formally, the marginal distributions should conform to an underlying PDE given by Fokker-Planck equation.
> >
> > Conceptually, using the forward-process diffusion loss will ensure that the training samples $x_t$ are correctly correlated with $x_0$. However, for reverse-process RL methods, they never involve the forward noising but only consider optimization on the reverse sampling trajectory $x_T\rightarrow\dots x_t\rightarrow x_{t-1}\rightarrow\dots\rightarrow x_0$. Therefore, there is no ensurence of forward consistency, because the intermediate $x_t$ could locate in any distribution, as long as the whole chain gives a high-reward final $x_0$. It is not guaranteed that $\pi_\theta(x_t)$ could be obtained by adding certain amount of noise to $x_0\sim\pi_\theta(x_0)$.

---

### Meta-Review · Area_Chair_NLpn · 2025-12-28

**Summary:**

The paper introduces Diffusion Negative-aware FineTuning (DiffusionNFT), a novel online reinforcement learning framework for diffusion and flow models. Unlike existing methods (e.g., DDPO, FlowGRPO) that discretize the reverse sampling process, often leading to solver restrictions and high computational costs. DiffusionNFT optimizes the policy directly on the forward process via flow matching. The method contrasts positive and negative generations to define an implicit policy improvement direction, incorporating reinforcement signals into a supervised learning objective. This formulation allows for the use of arbitrary black-box solvers (including efficient ODE solvers), eliminates the need to store full sampling trajectories, and removes the need for likelihood estimation. Empirical results demonstrate that DiffusionNFT is up to 25$\times$ more efficient than FlowGRPO while achieving superior or comparable performance on benchmarks like GenEval and OCR, and notably performs well without Classifier-Free Guidance (CFG).

**Reviewer Concerns:**

Addressed Concerns

- Missing Baselines/Models (Reviewer fF9P, w5yg): The reviewers requested comparisons to iterative DPO/KTO methods and newer models like FLUX, as well as concurrent work like DanceGRPO. The authors explained the theoretical limitations of likelihood-based DPO/KTO in this context, clarified that DanceGRPO is algorithmically similar to the baseline FlowGRPO, and provided new experimental results applying DiffusionNFT to FLUX, showing consistent improvements.

- Clarity and Notation (Reviewer fF9P): Concerns regarding the definitions of $o$, $p_{old}$, and the specific reward formulations were addressed with detailed explanations in the rebuttal, clarifying the probabilistic inference framing.

- Distribution Shift & Negative Samples (Reviewer niMD): The reviewer questioned how the method handles distribution shift and if negative samples are used efficiently. The authors clarified the role of the negative loss term in defining the "avoid" direction and explained the use of soft updates (EMA) for the data collection policy to mitigate catastrophic distribution shift.

- CFG Analogy (Reviewer w5yg): The confusion regarding "offline" CFG integration was clarified as a conceptual analogy rather than a training procedure.

Outstanding Concerns:
   - There are no significant outstanding concerns. The reviewers who engaged (fF9P, w5yg) gave high scores (8), and the concerns of niMD (distribution shift) were standard for RL papers and addressed via the EMA explanation.

**Reviewer Scores:**

I cannot reliably answer this counterfactual question without putting words in reviewers’ mouths. I will not impute score changes beyond what reviewers explicitly stated in the discussion. I instead provide a faithful synthesis of the discussion outcomes and remaining points of disagreement.

---

### Decision · Program_Chairs · 2026-01-26

Accept (Oral)